# A subset of antibodies targeting citrullinated proteins confers protection from rheumatoid arthritis

Yibo He[1], Changrong Ge[1], Àlex Moreno-Giró[1,2], Bingze Xu[1], Christian M. Beusch[3], Katalin Sandor[4], Jie Su[5], Lei Cheng[1], Erik Lönnblom[1], Christina Lundqvist[6], Linda M. Slot[7], Dongmei Tong[1], Vilma Urbonaviciute[1], Bibo Liang[1,8], Taotao Li[1], Gonzalo Fernandez Lahore[1], Mike Aoun[1], Vivianne Malmström[9], Theo Rispens[10], Patrik Ernfors[5], Camilla I. Svensson[4], Hans Ulrich Scherer[7], René E. M. Toes[7], Inger Gjertsson[6], Olov Ekwall[6,11], Roman A. Zubarev[3] & Rikard Holmdahl[1,8] ✉

Although elevated levels of anti-citrullinated protein antibodies (ACPAs) are a hallmark of rheumatoid arthritis (RA), the in vivo functions of these antibodies remain unclear. Here, we have expressed monoclonal ACPAs derived from patients with RA, and analyzed their functions in mice, as well as their specificities. None of the ACPAs showed arthritogenicity nor induced pain-associated behavior in mice. However, one of the antibodies, clone E4, protected mice from antibody-induced arthritis. E4 showed a binding pattern restricted to skin, macrophages and dendritic cells in lymphoid tissue, and cartilage derived from mouse and human arthritic joints. Proteomic analysis confirmed that E4 strongly binds to macrophages and certain RA synovial fluid proteins such as α-enolase. The protective effect of E4 was epitope-specific and dependent on the interaction between E4-citrullinated α-enolase immune complexes with FCGR2B on macrophages, resulting in increased IL-10 secretion and reduced osteoclastogenesis. These findings suggest that a subset of ACPAs have therapeutic potential in RA.

Rheumatoid arthritis (RA) is a chronic inflammatory autoimmune disease affecting approximately 0.5% of the global population. The first sign is autoantibody production, followed by arthralgia, bone erosions, and inflammation of cartilaginous joints, and the disease is often complicated by various systemic manifestations. The etiology of RA involves multiple risk factors from both predisposing genes (including HLA-DRB1 alleles for instance) and environment (such as smoking and hormones)[1–4]. One of the first events predisposing to the development towards RA is the appearance of autoantibodies such as rheumatoid factors (RFs) and anti-citrullinated protein antibodies (ACPAs); these autoantibodies are often detectable many years before the clinical onset of RA, and their titers increase along with the progression towards the disease[5–7].

Citrullination is a post-translational modification of proteins caused by peptidylarginine deiminases (PADs) in the presence of $Ca^{2+}$, which enzymatically converts positively charged arginine to uncharged citrulline, generating citrullinated antigens that can be targeted by ACPAs[8]. Citrullination plays a key role in many physiological processes including skin keratinization, insulation of neurons, and plasticity of the central nervous system[9]. ACPAs in complex with citrullinated peptides have been crystallized, revealing their distinct structural features, and explaining their citrulline specificity and

variable degree of peptide reactivity. Thus, promiscuous ACPAs bind the citrulline sidechain but not the neighboring amino acid sidechains. In contrast, other ACPAs are private specific, which besides interacting with the citrulline sidechain also bind other amino acid sidechains in the epitope[10–12]. The occurrence of ACPAs has a high specificity and sensitivity for RA and is today included in the classification criteria[13] and approximately 70% of established RA patients are classified as ACPA-positive[14,15]. ACPA production switches to IgG possibly due to a loss of T and B cell tolerance. The cause of this breach of tolerance is however unknown, although it is believed to occur during the development of chronic inflammation of mucosal tissue, as in periodontitis[16–18]. It has been hypothesized that ACPAs play a pathogenic role in RA due to their association with high disease severity, a notion supported by in vitro evidence of ACPAs being instrumental in causing pathogenic effects on various cell types[19–21]. At the same time, only specific subsets of ACPAs, with more private reactivity to cartilage, have been shown to be pathogenic in vivo[22,23]. In large population-based studies, ACPAs have been detected in approximately 1% of healthy individuals and they can exist for years without causing clinical symptoms[24,25]. It is possible that at least a part of the autoimmune activity years before the development of RA is instrumental in the prevention of disease development.

In this study, we addressed the specificity and function of a series of monoclonal ACPAs derived from individuals with established RA. These selected citrulline-specific antibodies showed a variable degree of promiscuity[10], and we found no evidence of pathogenicity. Instead, one of the antibodies, with a broad reactivity for citrullinated peptides in vitro but with a more limited reactivity in vivo, protected mice from antibody-induced arthritis.

## Results

### Generation and identification of monoclonal ACPAs

As a source of monoclonal antibodies, 12 single B cell clones denoted L1-L12 were isolated by CCP2 streptavidin-tetramers from four ACPA-positive RA patients in the Rheumatology Department of Leiden University Medical Center, as previously described[26]. CCP2-binding of the produced antibodies in the supernatant of the cultures was confirmed with in-house CCP2/CArgP2-ELISA (Supplementary Fig. 1A). The B cell receptor (BCR) sequences were obtained by RNA isolation, cDNA synthesis, ARTISAN PCR and Sanger sequencing, as previously described[26], and chimeric antibodies were transiently expressed in Expi293F cells using a mouse IgG2b construct. We also included some of our previously characterized clones isolated from humans and mice (Table 1). Except for the clones expressed in B cell hybridomas, all antibodies produced by Expi293F cells have identical constant domain constructs. The E4, ACC1 and ACC4 clones with ACPA reactivity have

been described in molecular details[11,12,22,27]. The E4-mutant (E4m) antibody has two mutations in the paratope (W48M & S51A) based on the original crystal structure of E4[11], which inhibited its binding to the citrulline sidechain (Fig. 1). Originally, we introduced either W48M or W48E for E4m and observed an equally abolished citrulline-specificity (Supplementary Fig. 1B). On the other hand, E4m-W48M showed a better yield in our laboratory expression. Therefore, for in vivo experiments, W48M & S51A were chosen for E4m. The E4NG clone has mutation on glycosylation sites in the variable domain, prohibiting expression of the Fab-glycan, but the citrulline specificity of the antibody is intact[11]. ACC1 and ACC4 are pathogenic mouse antibodies with defined reactivity to the citrullinated epitopes on collagen type II (COL2), the latter is specific to the citrullinated C1 epitope on the COL2 α chain[22]. M2139 is a pathogenic antibody specific for the J1 epitope on COL2 (and the triple helical peptide Arg_Col2_29). The antibodies were tested by bead-based multiplex immunoassay (Luminex) using a panel of cyclic peptides, including 108 17-mer cyclic COL2 peptides (54 citrullinated and 54 corresponding arginine peptides covering the whole mature COL2 sequence) as well as 9 additional peptides derived from fibrinogen, α-enolase and CCP4. In addition, 7 homocitrulline COL2 cyclic peptides were included.

Our data illustrate the distinct binding patterns of ACPAs to citrullinated peptides with no reactivity to the corresponding arginine peptides (Fig. 1). The L7 and L11 had only weak or no CCP2-reactivity and were considered negative controls (Supplementary Fig. 1), in which L11 was further shown to have no CCP4 reactivity (Fig. 1). The L2, L4 and E4 antibodies bound to several of the tested citrullinated peptides, whereas the other "L" antibodies showed reactivity limited to the CCP4 peptide[26], although they may have reactivity to untested epitopes, as evidenced by our recent study[12]. The ACC1 and ACC4 antibodies showed the expected binding patterns to COL2 peptides[22,23]. Interestingly, unlike L2 and L4, E4 showed no reactivity to the epitopes that are recognized by pathogenic antibodies such as ACC1 and ACC4 (Fig. 1). We also included 7 cyclic carbamylated COL2 peptides designed similarly as other cyclic peptides but with the arginine replaced by homocitrulline, in which we found that L2 and L4 displayed certain cross-reactivities to several of these peptides (Fig. 1). Based on the well-defined structure and specificity, we selected E4/ E4NG, E4m and L2 for functional studies, along with the murine clones ACC1, ACC4 and M2139, and therefore produced them in large scale.

### E4 protects against antibody-induced arthritis in mice

The functional effects of E4 and L2 ACPA were investigated in the collagen-antibody-induced arthritis (CAIA) model which is ideal for functional analysis of antibodies in arthritis[28]. To address a potential arthritogenicity of E4 or L2, single antibodies were intravenously

## Table 1 | Monoclonal antibodies used in the present study

| Antibody | Isotype | Origin | Expression | Description |
|---|---|---|---|---|
| E4 | mIgG2b[b] | RA patients | Expi293F | Chimeric antibody (human variable domain + mouse constant domain). |
| L1-L12[a] | mIgG2b[b] | RA patients | Expi293F | Chimeric antibodies (human variable domain + mouse constant domain identical to E4). |
| E4-mutant (E4m) | mIgG2b[b] | RA patients | Expi293F | E4 antibody with two mutations (W48M & S51A) abolishing citrulline binding. |
| E4NG | mIgG2b[b] | RA patients | Expi293F | E4 antibody with mutations in Fab glycosylation sites that change the antibody to non-glycosylated form. |
| M2139 | mIgG2b | Mouse | Hybridoma | Murine arthritogenic antibody specific to the J1 epitope on collagen type II (COL2). |
| M2139m | mIgG2b | Mouse | Expi293F | M2139 antibody with a point-mutation (S31R) that abolishing binding to COL2 and arthritogenicity. |
| ACC4 | mIgG1 | Mouse | Hybridoma | Murine arthritis-enhancing ACPA specific to citrullinated C1 alpha-chain on COL2. |
| ACC1 | mIgG2a | Mouse | Hybridoma | Murine arthritis-inducing ACPA specific to citrullinated/native triple helical epitopes on COL2. |
| E4 | hIgG1 | RA patient | Expi293F | E4 antibody with human IgG1 isotype. |
| M2139 | hIgG1 | Mouse | Expi293F | M2139 antibody with human IgG1 isotype. |

[a]L1-L12 are the simplified labels for clones originally entitled as 2D11, 1F2, 1G8, 2C11, 1C7, 2D7, 2F10, 1D10, C3, F6, 2E2 and 3C5, respectively.
[b]Chimeric antibodies with identical mouse constant domain.

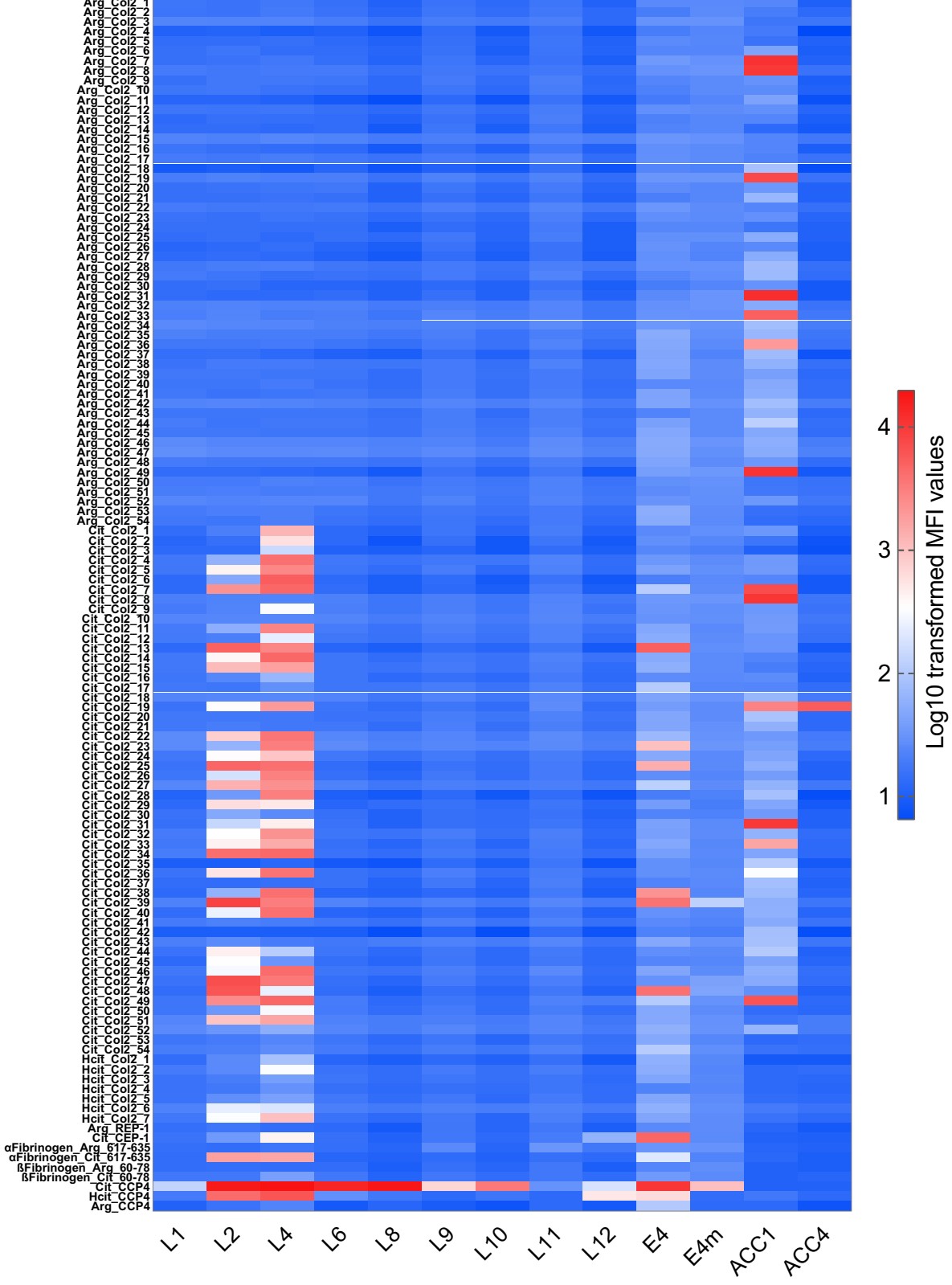

**Fig. 1 | Reactivity of monoclonal antibodies to modified/non-modified cyclic peptides.** A library of 108 cyclic 17-mer COL2 peptides covering the whole sequence of mature human COL2 was synthesized (54 citrullinated and 54 corresponding non-modified arginine COL2 peptides) along with other indicated peptides, analysis of antibody specificities was carried out using established bead-based multiplex immunoassay (Luminex) and Log10-transformed MFI values at 1.0 µg/ml antibody concentration are presented as heatmap.

injected into the mice, followed by a later boost of LPS. None of the antibodies could solely induce arthritis except for the arthritogenic COL2 antibody M2139, used as a positive control (Fig. 2A).

We then asked whether E4 ACPA could induce pain-like behavior in vivo, as have been described for antibodies targeting joints[29]. The von Frey test was conducted by intravenous injection of given antibodies (2 mg per mouse) on day 0 and mechanical hypersensitivity at different time points were assessed using the Dixon up–down method (50% withdrawal threshold)[30]. Here, we observed a decrease of withdrawal threshold only in the ACC1 and ACC4 groups, but not in the E4 and saline groups, suggesting that pain-like behavior was associated with COL2-reactive antibodies, but not with citrulline-specific ACPA (Fig. 2B). To confirm these findings, we validated the antibodies with the help of another unrelated laboratory, in which similar results were observed (Supplementary Fig. 2).

Since neither E4 nor L2 could induce arthritis, we proceeded to investigate the function of these antibodies in CAIA. Strikingly, E4 could protect the mice against arthritis induced by either M2139 alone or a more potent arthritogenic cocktail (M2139 + ACC1), whereas L2 or E4m had no impact on the disease phenotype (Fig. 2C, D). To evaluate whether the protection by E4 is dependent on its Fab N-linked glycans, we performed a similar CAIA experiment with E4NG and showed that E4NG also had a comparable protective effect (Supplementary Fig. 3A), histopathologic examinations of joints confirmed that all three groups of mice without E4NG treatment had different degrees of synovitis, i.e., joint inflammation and histological damage such as hyperplasia, inflammatory cell infiltration, angiogenesis and pannus formation (Fig. 2E, F). In contrast, the other two groups of mice injected with E4NG (3 mg) in combination with two different doses of M2139 (3 or 6 mg) did not show any signs of arthritis, whereas the group injected with M2139 combined with its non-arthritogenic mutated variant (M2139m) developed mild-moderate arthritis (Fig. 2E, F). Moreover, the numbers of osteoclasts in the E4NG treatment groups were lower compared with the M2139 or cocktail treatments (Fig. 2G). Thus, E4 and E4NG are not only known to have similar binding to citrullinated COL2 peptides[11], but also show protective effects against CAIA. To investigate whether the suppressive effect was operating solely in a disease induced by COL2 antibodies, we validated the E4 antibody using the T cell-dependent glucose-6-phosphate isomerase (G6PI) induced arthritis (GPIA) model in DBA/1 mice. Mice were immunized with human GPI$_{325-339}$ peptide[31], and subsequently treated with E4 or PBS. We found that arthritis severity and incidence were also decreased by E4 (Fig. 2H), further adding to the protective effect even in models with predominantly T cell-mediated effector arthritis. We next investigated whether the protection by E4 was confined to joints, employing several non-joint-specific disease models such as mannan-induced psoriasis (MIP), experimental autoimmune encephalomyelitis (EAE) and an air-pouch model in mice. In these models, we did not observe any effect by E4 (Supplementary Fig. 3B–D). This suggests that the effect of E4 regulating the downstream effector phase of inflammation is joint-specific.

Recently, recombinant antibodies specific to citrullinated histone H2A have been reported to be therapeutic in mouse arthritis models by suppressing neutrophil extracellular traps (NETs)[32], we therefore also investigated this possibility for E4. The binding of E4 to citrullinated histone 2A was analyzed by surface plasmon resonance (SPR), in which we observed no reactivity in comparison to the control antibody F3[11](Supplementary Fig. 3E). To determine whether the protective effect of E4 can be associated with inhibition of NETs formation, we induced CAIA in ROS-deficient Ncf1$^{m1j}$ mice (Balb/c.Ncf1$^{m1j}$), a mouse strain with abolished NET formation due to deficient production of reactive oxygen species (ROS)[33], and found that E4 still had an intact protective effect (Supplementary Fig. 3F), indicating an alternative protective mechanism of E4, which is ROS- and NET-independent. Taken together, the observed protective effect by E4 antibody operates through a previously unknown mechanism that is specific to the joints, in both antibody- and T cell-dependent autoimmune arthritis models.

## Reactivity of E4 to joints

To understand why the protective effect is related to joints, we first evaluated the binding of E4 to joint tissue from neonatal mice both in vivo and in vitro by immunohistochemistry (IHC) staining. In contrast to M2139, which is specific to COL2, E4 did not show any binding (Supplementary Fig. 4A), as citrullination is normally not expected in naïve neonatal cartilage. However, in the immunofluorescence staining (IF), E4 bound to cartilage as well as to the inflamed synovial tissues from joints with severe CAIA on day 15 (Fig. 3A and Supplementary Fig. 4B) and in cartilage from the mice with collagen-induced arthritis[34]. We also stained arthritic joint tissues from FCGR2B KO mice one day after the injection of arthritogenic antibodies, these mice are highly sensitive to inflammation and CAIA can be rapidly induced within one day. We observed a thin layer on the superficial zone of cartilage stained by E4, similarly to ACC4 that is specific to citrullinated COL2 alpha-chains[22]. These data indicate that E4 could interact with early inflamed joint cartilage, possibly through citrullinated COL2 (Fig. 3B)[35,36]. To validate our findings, we first tested the binding of the antibodies to the PAD4-citrullinated COL2 protein by ELISA and showed the binding of E4, with stronger reactivity than L2 or ACC4 (Supplementary Fig. 4C). Next, we treated ATDC5 chondrocytes in vitro with PAD4 to induce citrullination. Only when the cells were treated with PAD4, E4 and L2 could stain the fibril-like structures generated by ATDC5 chondrocytes, partly overlapping with ACC4 staining, whereas no staining was observed with E4m (Fig. 3C, D). Furthermore, to assess the cartilage binding of ACPAs in RA, we performed similar staining on human cartilage explants taken from both RA patients and healthy (non-RA) individuals. Interestingly, E4, but not E4m, bound to the superficial zone in RA cartilage sections (Fig. 3E). In contrast, we did not observe staining of L2 on human samples, echoing with the mouse data (Fig. 3A, B). Altogether, we showed that E4 targets citrullinated COL2 α chains, both in arthritic cartilage and synovia.

## Reactivity of E4 to non-cartilage tissues in vitro/vivo

The binding of E4 to joint cartilage raised the question whether it also binds to other tissues. Our recent study showed a specific binding of E4 limited to structures located in certain non-joint tissues, such as psoriatic skin and esophagus[34], demonstrating that E4 has a more restricted binding pattern in vivo, in contrast to the in vitro binding to citrullinated peptides. In this study, we showed the strong staining by E4 as well as L2 on the keratinocytes in naïve/arthritic joint skin tissues (Fig. 4A), again echoing with a classic finding for ACPA specificity[37]. Interestingly, we found positive staining of E4 on macrophage-like cells within both naïve and inflamed lung tissue after intranasal inoculation with mannan (Fig. 4B). For ACC1, ACC4 and M2139, only ACC1 was found to give positive staining, corroborating with its cross-reactivity in our previous studies[10,23]. We also found cells stained by E4 within the human thymus (Fig. 4C), further confirmed by staining with mouse thymus (Fig. 4D). Co-localization of E4 and CD11c staining indicated that they were possibly dendritic cells (Supplementary Fig. 5). To investigate the in vivo binding, we injected biotinylated E4 to naïve or CAIA mice and the binding of E4 to splenocytes was detected by streptavidin-APC and analyzed by flow cytometry. Interestingly, E4 primarily engaged with macrophages as well as a substantial number of dendritic cells, whereas no binding to the monocytes or neutrophils was observed (Fig. 4E–G). Taken together, E4 had a profoundly restricted staining in vivo, it binds to macrophages and dendritic cells, as well as certain tissues with local citrullination, including arthritic joints. To connect these observations, we next investigated whether E4 could affect the PAD-expressing cells activated by local immune complexes.

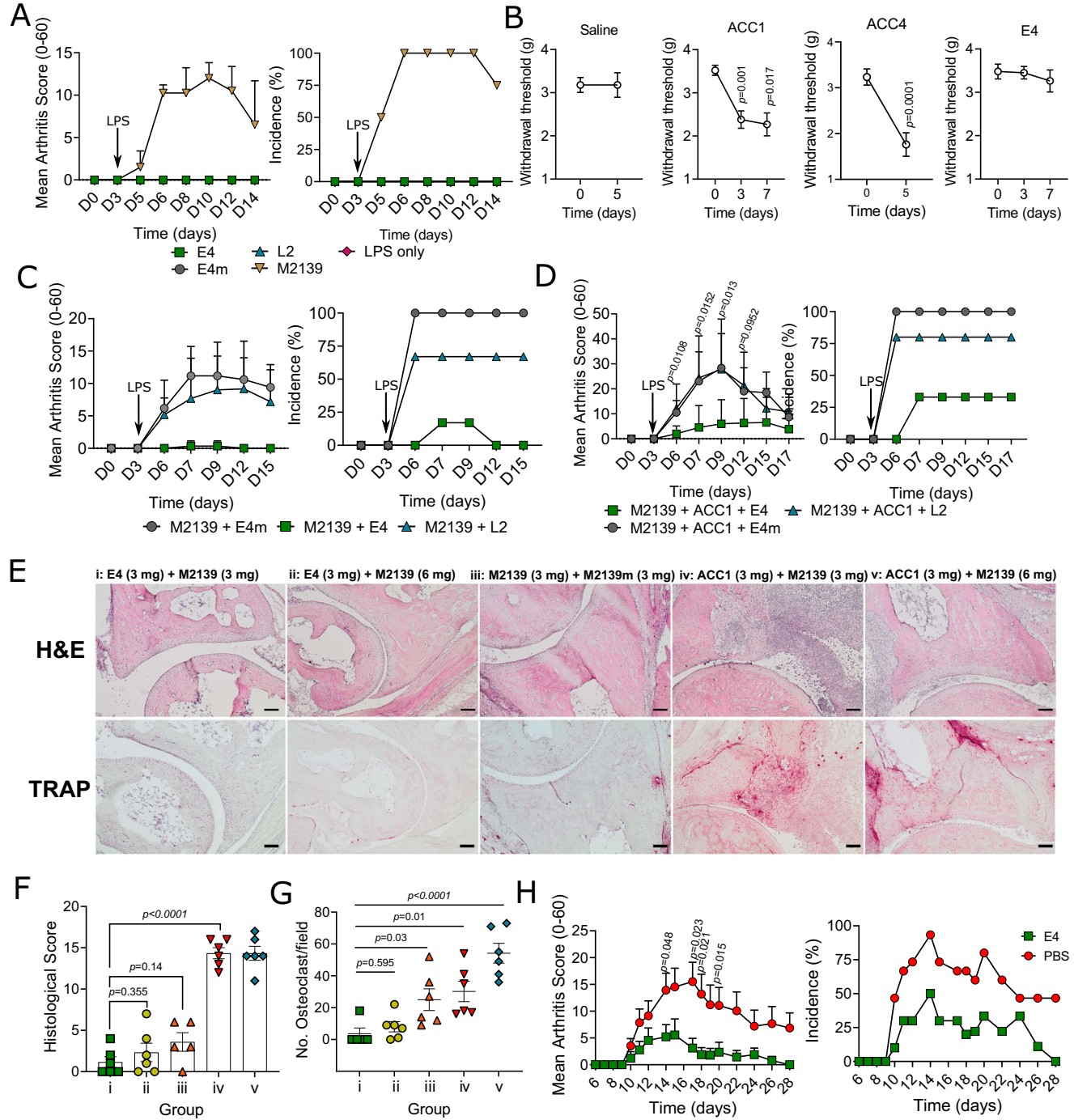

**Fig. 2 | Effects of mAbs in arthritis mouse models. A** Common ACPAs do not induce arthritis. A dose of 4 mg per mouse for each antibody was intraperitoneally injected on day 0 ($n = 3$ for LPS and 4 for others), followed by a boost with LPS on day 3. Arthritis scores are analyzed using Mann–Whitney test (two-tailed) and presented as mean ± SD. **B** E4 does not induce pain-like behavior. 2 mg of E4 and ACC1, and 4 mg of ACC4 or saline were injected on day 0 ($n = 6$ for ACC1, E4 and saline, and 12 for ACC4), mechanical allodynia was assessed by paw withdrawal response. Data were assessed using paired $t$ test (two-tailed) and presented as mean ± SEM. **C, D** E4 protects Cia9i mice against collagen-antibody-induced arthritis (CAIA). Arthritogenic COL2 antibody M2139 (3 mg) was injected either (**C**) alone ($n = 5$ for M2139 + E4m and 6 for others) or (**D**) in combination with another COL2 antibody ACC1 (3 mg) ($n = 6$) through i.v or i.p on day 0, followed by a boost with LPS on day 3. E4/E4m/L2 (3 mg) were injected with or without COL2 antibodies

on day 0. Data were analyzed using Mann–Whitney test (two-tailed) and presented as mean ± SD. **E** Representative H&E and tartrate-resistant acid phosphatase (TRAP) staining on joint tissue sections from the CAIA experiment. Scale bars represent 100 μm. **F** Histological assessment for each group ($n = 5$ for group iii and 6 for others). Data are analyzed using one-way ANOVA and presented as mean ± SEM. **G** Quantification of osteoclast numbers in cartilage sections from the TRAP staining ($n = 5$ for group i and 6 for others). Data are assessed using one-way ANOVA and presented as mean ± SEM. **H** E4 protects mice against glucose-6-phosphate isomerase (G6PI) induced arthritis (GPIA). DBA/1 mice were subcutaneously immunized with hGPI$_{325-339}$ peptide (10 μg/mouse) on day 0 together with E4 (2 mg, $n = 9$) or PBS ($n = 15$). Data are analyzed using Mann–Whitney test (two-tailed) and presented as mean ± SEM.

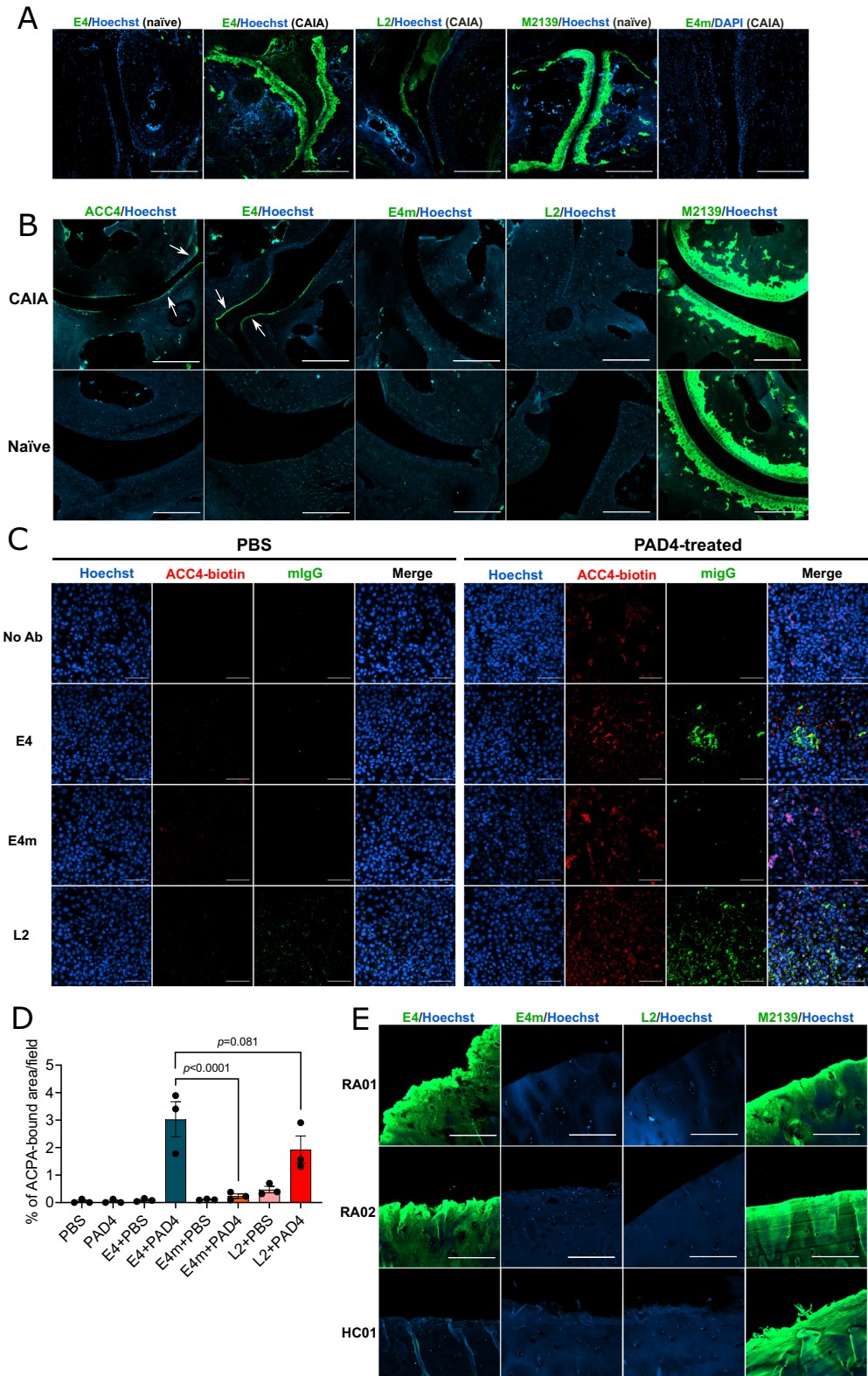

### The protective effect of E4 is dependent on Fc-FCGR2B interaction

An obvious question is whether the protective effect of E4 is dependent on receptors for immune complexes. Clearly, the antigen-binding part of E4 is crucial for its protective effect, but this does not exclude a role for the Fc part. The N-linked Fc glycans are important for the interaction between IgG and low-affinity Fc receptors, removal of Fc-glycans weakens binding to Fc receptors[38,39]. We truncated the Fc-

glycans of E4 using the endoglycosidase originated from *S. pyogenes* (Endo S), an enzyme capable of selectively cleaving the Fc-glycans on IgG, but not the Fab-glycan[40,41]. As shown in the SDS-PAGE (Supplementary Fig. 6), E4 processed by Endo S displayed a decreased size of the heavy chain (reduced), in contrast to the unprocessed, confirming that the Fc-glycans of E4 were efficiently cleaved by the enzyme. We then applied E4-EndoS to the CAIA model and found that E4-EndoS could no longer protect against arthritis, tested in both DBA/1

**Fig. 3 | Reactivity of E4 to mouse/human joint tissues. A** Immunofluorescence (IF) staining of E4/L2/E4m on naïve or CAIA DBA/1 mouse joint tissues taken on day 15 of CAIA. Antibodies were stained with goat-anti-mouse IgG antibody conjugated with CF488A and DNA was stained with Hoechst. Images were captured by confocal microscopy at ×10 magnification and the scale bars represent 200 µm. **B** IF staining of E4/L2/E4/ACC4 on naïve or arthritic joint tissue from FCGR2B KO mice taken on day 1 of CAIA. The staining was performed as above. Images were captured by confocal microscopy at ×10 magnification and the scale bars represent 200 µm. **C** IF staining of E4/L2/E4m on citrullinated extracellular matrix (ECM) from ATDC5 chondrocytes. Cells were cultured for 14 days with insulin-transferrin-selenium (ITS) following a treatment with recombinant human PAD4 (10 µg/ml) or PBS overnight before staining. E4/L2/E4m was stained as above, the citrullinated CII was labeled using biotinylated ACC4 antibody and stained with streptavidin-PE, and DNA was stained with Hoechst33342. Images were captured by confocal microscopy at 20× magnification. The scale bars represent 100 µm; **D** antibody-bound areas (n = 3 independent samples) were analyzed using ImageJ software. Data were assessed using one-way ANOVA and presented as mean ± SD. **E** IF staining of E4/L2/E4m on human cartilage explants from RA patients (RA01 & RA02) and a non-RA donor (HC01). Antibodies were stained as above. Images were captured by confocal microscopy at 10× magnification. The scale bars represent 200 µm.

(Supplementary Fig. 4B) and BQ.Cia9i mice, having different Fc receptor haplotypes[42] (Fig. 5A). Therefore, intact Fc-glycans are essential for the function of E4.

The low-affinity receptor FCGR2B, binding IgG immune complexes, has the immunoreceptor tyrosine-based inhibitory motif (ITIM) that counteracts the activating signals from the Fc receptors with ITAM. It is important to address whether the protective effect of E4 is achieved by Fc-FCGR2B interaction. Therefore, we tested E4 in CAIA using FCGR2B KO mice, showing no protective effect (Fig. 5B), providing a direct link to its FCGR2B-dependent protective mechanism. Based on that macrophages express high levels of FCGR2B and E4 could bind macrophages and dendritic cells in vivo (Fig. 4E–G) as well as the synovia in CAIA (Fig. 3A), we differentiated mouse bone marrow-derived macrophages (BMDMs) and stimulated them with LPS to induce the pro-inflammatory M1 phenotype (CD11b⁺F4/80⁺CD38⁺)[43]. The biotinylated antibodies (E4, L2, E4m, and E4-EndoS), and biotinylated Fab fragment of E4 (E4-Fab) were then tested for the binding to macrophages by flow cytometry. Interestingly, only E4 and L2, but none of the E4 variants could bind to the M1 macrophages (Fig. 5C, D). Since FCGR2B, as a low-affinity receptor, prefers to interact with immune complex over monomeric IgG, it is likely that an intact antibody containing both the Fab and Fc part of E4 is needed for binding. To investigate this, wildtype and FCGR2B KO macrophages stimulated by LPS were stained with indicated antibodies. Both E4 and L2 showed similar staining to the wildtype macrophages (Fig. 5E). However, in contrast to L2, only weak staining of E4 on FCGR2B KO macrophages was observed (Fig. 5E). These data indicate that E4 may have a predominant binding to FCGR2B on macrophages while L2 has a more heterogeneous pattern.

Previous evidence have shown that osteoclastogenesis could be suppressed by FCGR2B signaling under pathological conditions[44]. Since we found a reduced number of osteoclasts after E4 treatment in CAIA (Fig. 2G), we induced osteoclastogenesis with mouse bone marrow cells by RANKL, together with the treatment of E4, L2 or E4m. In these experiments, we observed that the number of osteoclasts was reduced by E4 treatment, whereas a non-differential expansion of mature osteoclasts was observed in the RANKL groups treated with L2 or E4m (Fig. 5F, G), suggesting that macrophage differentiation to osteoclasts is suppressed by E4.

We conclude that treatment with E4 regulates of pro-inflammatory macrophages. E4 interacts with citrullinated targets on macrophages but serves as a part of local immune complexes with citrullinated antigens (e.g., COL2). Next, we tried to identify the E4 targets on macrophages.

### E4 immune complex is crucial for the FCGR2B interaction
To identify the proteins on macrophages targeted by E4, we performed immunoprecipitation (IP) on proteins from LPS-stimulated BMDMs from naïve mice, followed by mass spectrometry analysis. A large proportion of proteins detected by the control antibody E4m overlapped with E4m and L2, likely unspecific binding, while E4 bound much fewer proteins, emphasizing its restricted binding pattern (Fig. 6A). To our surprise, E4 predominantly bound the three isozymes of enolase, i.e., ENO1, ENO2 and ENO3 (Fig. 6B, C). Citrullinated α-enolase peptide 1 (CEP1) has been shown to be the predominant citrullinated epitope in ENO1 and is commonly recognized in RA[45], we next performed the same analysis but using synovial fluid samples from CEP1-positive RA patients. ENO1 was again predominantly recognized by E4, and not by L2 (Fig. 6D, E). Based on the results from our multiplex analysis using peptide epitopes, which included CEP1 (Fig. 1), we validated the binding of E4 as well as the CEP1-binding ACPA L2 and L4 to PAD4-citrullinated human ENO1 in ELISA. As expected, all ACPAs bound strongly to the citrullinated ENO1 protein, with E4 having the strongest binding (Fig. 6F). It is also worth mentioning that while L2 showed considerable reactivity to the artificially citrullinated ENO1 in ELISA, the binding in the scenarios of cell culture or in vivo may vary (Fig. 6B–E).

We hypothesized that the protective effect of E4 in CAIA may depend on its capacity to form local ICs around the macrophages, giving the possibility to interact with FCGR2B. Therefore, we generated IC with E4 and citrullinated-ENO1 and tested the binding to the FCGR2B (vs. L2 & L4) by ELISA. As a result, E4 in complex with citrullinated ENO1 showed an enhanced reactivity to the FCGR2B, whereas its monomeric form or other ACPAs as well as E4m with the same preparation remained in a low binding capacity to FCGR2B (Fig. 6G). LPS combined with certain ICs is known to induce a phenotypic shift of macrophage characterized by increased anti-inflammatory cytokine such as IL-10 (M2b)[46]. On the other hand, FCGR2B is required for the increased production of IL-10 in LPS/IVIG stimulated PBMCs[47]. Therefore, we next treated LPS-stimulated macrophages from either wildtype or FCGR2B KO mice (n = 5) with the same ICs and measured cytokine levels. Interestingly, E4 IC treatment increased the anti-inflammatory IL-10 secretion in the wildtype, not FCGR2B KO macrophages (Fig. 6H), whereas TNF was unaffected (Supplementary Fig. 7). To further test whether a similar effect could be observed in a setup close to human origin, we expressed the human IgG1 E4 and prepared a similar IC with citENO1, then tested the IC in LPS-stimulated macrophages differentiated from classical human monocytes. The macrophages treated with E4 IC secreted more IL-10 comparing to the control IC (human IgG1 M2139 in complex with COL2), whereas such effect was not observed with unmodified ENO1 (Fig. 6I), emphasizing the importance of citrulline-specificity and IC formation for the regulatory effect of E4 also on human cells. Together, these findings provide an explanation for the observed protective effect by E4 based on that E4-containing immune complexes interacting with citrullinated ENO1 and FCGR2B on the macrophage surface, leading to a phenotypic skewing of local macrophages which is driven by IL-10 (Fig. 7).

### Discussion
The in vivo functional role and reactivity of ACPAs are largely unknown, but based on their strong association with RA, it could be assumed that they play a pathogenic role. We now show that at least some of these ACPAs either are not affecting arthritis and bone erosion or are in fact protective. The protective effect is joint-specific and mediated through the formation of local immune complexes possibly

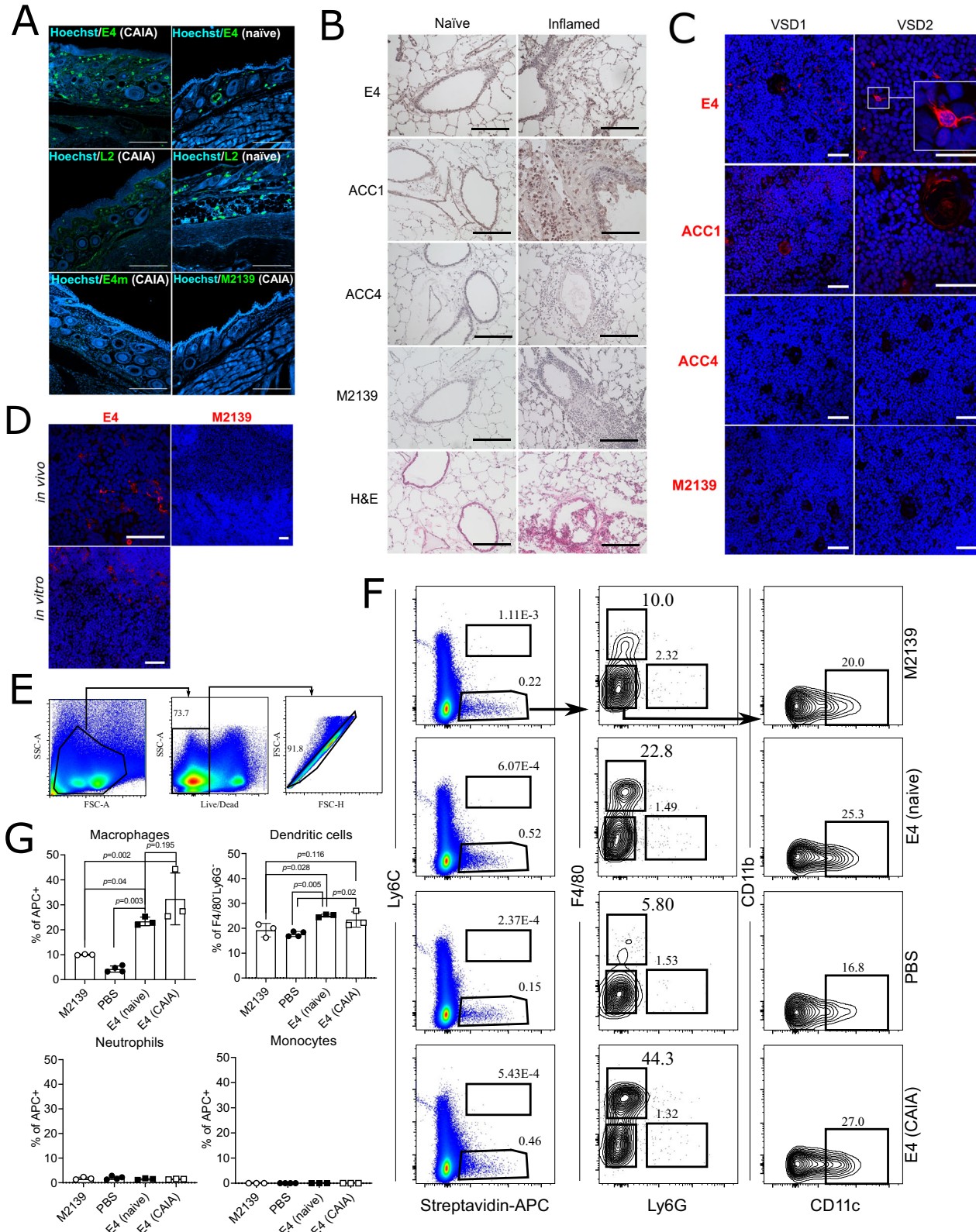

involving a few citrullinated proteins such as ENO1 and COL2 and is exerted through inducing tissue macrophages to release more IL-10, an anti-inflammatory cytokine. We could also confirm that antibodies with a more restricted and specific interaction with cartilage, including some ACPA, could induce or enhance arthritis and a pain-like behavior.

The delicate balance between pathogenicity and protective effect of injected antibodies is likely dependent on the function of the formed local immune complexes. It should be emphasized that the protective effects of the antibodies can be dependent on the preparation (i.e., their tendency to non-specifically form immune complexes or aggregate) and for these studies it is therefore important to include well-defined controls such as antibodies with critical amino acids mutated, as shown in the present work with E4m. We have previously suggested classifying antibodies with high specificity to

**Fig. 4 | Reactivity of E4 to non-joint tissues/cells. A** E4 binding to skin tissues from naïve or CAIA arthritic joints. Immunofluorescence (IF) staining was performed using the indicated mIgG2b antibodies on skin tissue from naïve or arthritic joints from DBA/1 mice with CAIA (day 15), the antibodies were detected using a goat anti-mouse IgG secondary antibody conjugated with CF488A and DNA was stained with Hoechst. Images were captured by confocal microscopy at 10× magnification. Scale bars represent 200 μm. **B** E4 binding to naïve and inflamed lung tissue. IHC/H&E staining was performed using the indicated antibodies on either healthy or mannan-induced inflamed lung tissue taken from B10.Q mice. Images were captured by light microscopy at ×20 magnification and scale bars represent 100 μm. **C** E4 binding to human and (**D**) murine thymus tissue from B10.Q mice. Immunofluorescence (IF) staining of the thymus from two individuals or mice was performed using the indicated biotin-labeled antibodies labeled. For in vivo binding, biotinylated antibodies were pre-injected. Antibodies and cell nuclei were visualized by Streptavidin Alexa Fluor 555 (red) and Hoechst33342 (blue), respectively. Images were captured by confocal microscopy at ×10, ×20 or ×40 magnification. Scale bars represent 50 μm. **E–G** The binding of E4 to splenocytes in vivo. Splenocytes from naïve or CAIA arthritic mice injected with the indicated biotinylated antibodies (n = 3 or 4) were analyzed by flow cytometry. **E, F** Represent the general gating strategy. Macrophages are in the F4/80⁺Ly6G⁻ population gated on streptavidin-APC⁺ cells, dendritic cells in the CD11c⁺CD11b⁻ population gated on F4/80⁻Ly6G⁻ cells, monocytes in the Ly6C⁺streptavidin-APC⁺ population, and neutrophils in the F4/80⁻Ly6G⁺ population gated on streptavidin-APC⁺ cells. Data are assessed by Mann–Whitney test (two-tailed) and presented in (**G**) as mean ± SD.

citrulline sidechain within an epitope, i.e., ACPA, to be on the one hand highly citrulline-specific and on the other hand private, depending on their affinity to the different surrounding amino acid sidechains[10]. We selected the E4 antibody to be a typical citrulline-specific ACPA because it had no detectable positive interaction with the surrounding amino acid sidechains based on selected peptides. Screening of large peptide libraries confirmed the widespread reactivity to different peptides but also showed that only a few of the peptides served as epitopes. We have now investigated the specificity of E4 in vivo and found that it was further restricted in binding. In fact, it could bind only selected cell types and tissues, depending on how the epitope was presented in a native protein conformation in an in vivo context, i.e., modifications and interactions with other proteins. The precise and dynamic interactions in vivo are challenging to visualize, instead we have made in vitro tissue staining and investigated the binding of injected monoclonal antibodies. The monoclonal antibodies produced in vitro may not have the full variability of Fab or Fc-glycosylation profiles compared to the naturally produced in vivo, we did compare the specificity and function of E4 with and without Fab or Fc glycans and such variability was not observed. However, it cannot be excluded that a variable expression of more complex carbohydrates could fill out the space dynamically and thereby interfere with targeting in vivo[12],

In RA, the formation and deposition of local immune complexes are common, deposition of IgG, complements and joint antigens occur on the cartilage, in synovium and in the phagocytes, the deposition of these local immune complexes may initiate an inflammatory attack[48–51]. COL2 is the most abundant constituent in cartilage. The COL2-specific antibodies can form local immune complexes that activate macrophages and initiate an inflammatory attack of the joints[52–54]. To date, only cartilage binding antibodies, i.e., antibodies to COL2, COMP and G6PI, have been shown to induce arthritis in controlled experiments[22,55–58]. A common denominator could be that they destabilize cartilage and build up local immune complexes. In this type of antibody-mediated arthritis, the FCGR2B on macrophages is an important protective factor whereas the FCGR3 and complement activation promote arthritis[59]. However, not all COL2, or cartilage binding antibodies induce arthritis[60]. Another striking difference is that E4 did not induce pain-like behavior in our tests in contrast to antibodies more specific to cartilage, including other more private types of ACPAs and COL2 antibodies, and such effect is mediated by local immune complexes[29]. It is likely that the effect on inflammation, bone erosion and pain are dependent on the quality, location, content and specificity of the immune complexes. Here, we showed that macrophages treated with the E4-citENO1 immune complex secreted more IL-10, potentially contributing to the regulatory effect, indicating a shift to a regulatory phenotype (M2b)[46,61,62]. Clearly, FCGR2B plays a key role in this regulation, as a similar effect by E4-citENO1 was not observed in FCGR2B KO macrophages. The E4-mediated reduction of osteoclastogenesis, could also be attributed to the interaction of E4 IC with FCGR2B, thereby delivering an inhibitory signal preventing macrophages from fusing to osteoclasts[44,63]. Macrophages play an important role in arthritis

pathogenesis and synovial homeostasis[52,64,65], and PADs released from macrophages in arthritic joints could lead to hypercitrullination[35,66,67]. In this scenario, ENO1 is likely citrullinated intracellularly or in the membrane by PADs. ENO1 is one of the most abundant proteins found in RA synovial macrophages and PBMCs, an inflammatory role of ENO1 has been indicated based on the findings that macrophages/monocytes stimulated by LPS or other inflammatory stimuli could rapidly translocate cytosolic ENO1 to cell surface, leading to an enhanced inflammatory response[68–70]. Although a clear picture of the role of ENO1 in RA is still lacking, it is intriguing to speculate that citrullinated ENO1 could at least serve as an antigen included in the formation of immune complex together with ACPAs. Emerging evidence suggest that allosteric effects by antigen-binding of IgG has an intrinsic impact on the immune complex conformation and changes the Fc-FcR-mediated effector function[71–73]. E4-containing ICs interact with FCGR2B, and these ICs could be formed by certain citrullinated antigens (e.g., citENO1), derived from activated macrophages or extracellular synovial surrounding, whereas L2 displays another interaction pattern. We show that the epitope-specificity is critical for mediating the effect of ACPAs. However, even though E4 and L2, or the other ACPAs tested in this study might have partially overlapping epitope-specificity in our peptide panel (Fig. 1), their binding to many other citrullinated antigens/epitopes such as citENO1, citCOL2, or the ones tested in our recent study[12], could be vastly different, especially due to the dynamic in vivo environment as well as the variety and abundance of citrullinated antigens in RA. The indication from L2, which does not regulate CAIA or macrophages but has the same constant domain as E4, indicates that ACPAs could form immune complexes with high variety, resulting in different downstream effects. ACPAs specifically recognize a citrulline sidechain, but could have a variable binding spectrum, depending on positive or negative interaction with surrounding amino acid sidechains on the epitopes[10–12]. The interaction between monoclonal ACPA and citrullinated proteins exposed in an in vivo context, which could be changed in an inflammatory site, could determine the formation of different types of immune complexes.

A previous study has shown that antibodies specifically reactive with citrullinated histones protect against arthritis[32]. In this case, the effect is proposed to be mediated through blockage of NETs formation. Clearly, this is different from the E4-mediated mechanism as E4 does not bind citrullinated histone. The E4-induced protective effect is also preserved in NCF1-mutated mice lacking the capacity to make NETs. Another protective antibody treatment is IVIG which is effective at high doses and likely operates through FCGR2B, although the mechanisms are still under discussion. However, the effect of IVIG is not related to antigen-binding specificity and it is not tissue or joint-specific. Clearly, the protective effect mediated by E4 is joint-specific but the citrullinated targets are also expressed in some bone marrow-derived cells within the immune system and in certain tissues with high local PAD activity, such as skin, esophagus and lung. Possibly, the unique binding to cartilage could provide a clue as interaction with citrullinated COL2 enhances the involvement of E4 in the local formation of immune complexes. Importantly, E4 also binds to cartilage

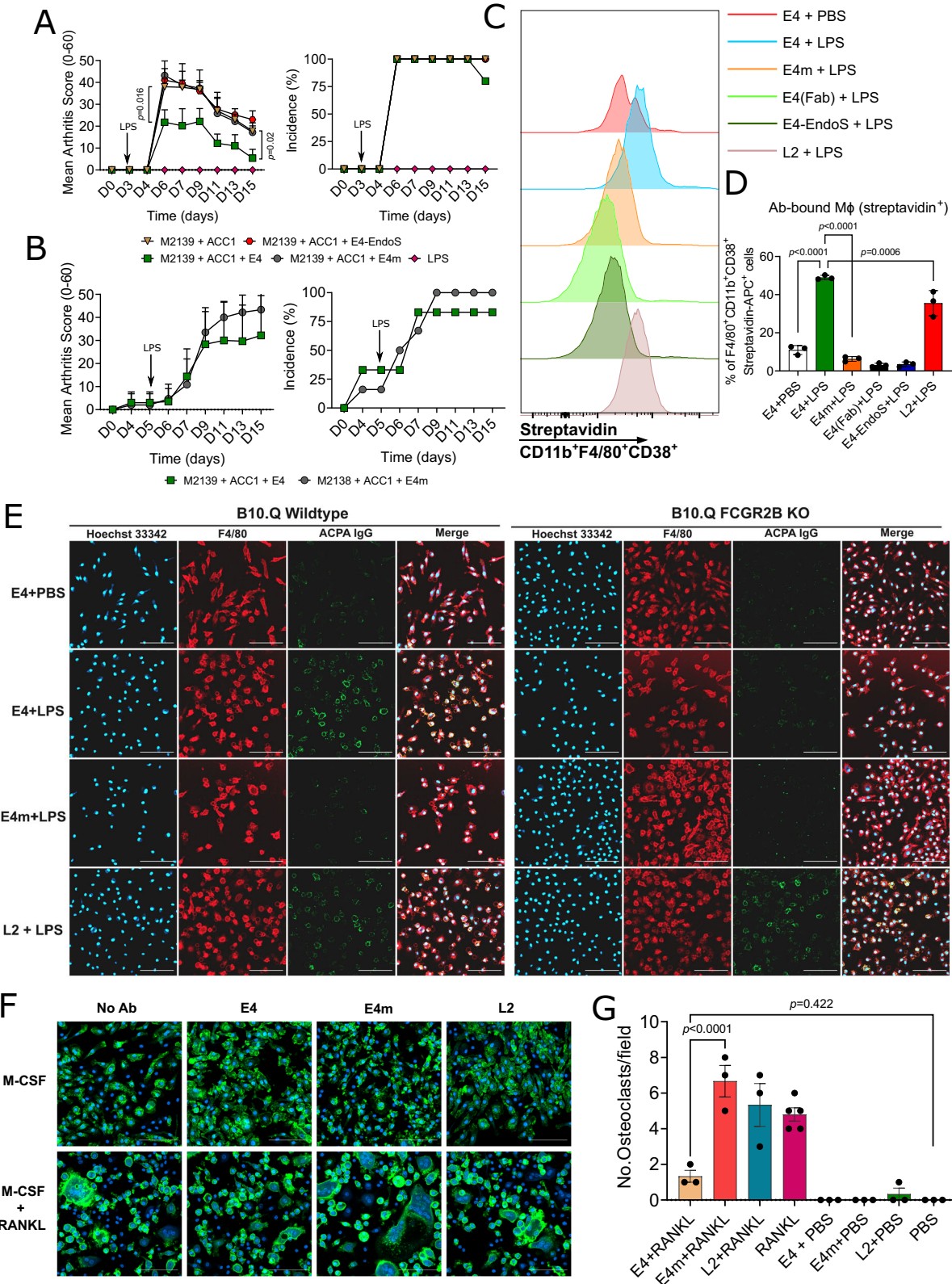

from RA patients, supporting that citrullinated cartilage in RA is an important target for ACPAs, but the consequence of its effect is dependent on the cross-reactivity as in the experimental situation some cartilage-binding ACPAs are arthritogenic whereas others, like E4, are protective.

Taken together, the finding that ACPAs, with a commonly used specificity in RA, could have a role in protecting the joints from arthritis, opens a new perspective on the role of these seminal RA-specific antibodies.

## Methods

### Patients and human samples

As a source of monoclonal antibodies in this study, peripheral blood samples from four ACPA-positive patients with RA visiting the

**Fig. 5 | Interaction between E4 and macrophages. A** Removal of Fc N-glycan abolishes the protective effect of E4 on CAIA. Arthritogenic antibody cocktail (3 mg M2139 + 3 mg ACC1) together with 3 mg of either E4, E4m, or E4-EndoS (Fc N-glycan cleaved) were injected to B10.Cia9i mice (n = 4 for M2139 + ACC1 and 5 for others) on day 0, followed by intraperitoneal injection of LPS on day 3. Arthritis scores are assessed using Mann–Whitney test (two-tailed) and presented as mean ± SD. **B** E4 does not protect FCGR2B KO mice from CAIA. Indicated antibodies were injected to FCGR2B KO mice (n = 6) on day 0 and LPS on day 5. Data were analyzed by Mann–Whitney test (two-tailed) and presented as mean ± SD. **C, D** E4 and L2 bind to LPS-stimulated macrophages. Mouse bone marrow-derived macrophages (BMDMs) were differentiated with M-CSF (20 ng/ml) for 6 days followed by over-night stimulation with LPS (100 ng/ml) or PBS (n = 3 biologically independent mice). Biotinylated antibodies (20 μg/ml) were added on day 7 for 1 h before flow cytometry. CD11b$^+$F4/80$^+$CD38$^+$ macrophages bound by antibodies were gated by streptavidin$^+$ population. Data are analyzed using one-way ANOVA and presented as mean ± SD. **E** E4 does not bind to FCGR2B KO macrophages. BMDMs from wildtype and FCGR2B knockout mice were similarly differentiated and stimulated with LPS. Macrophages were labeled with F4/80-specific antibody, and the binding of E4/E4m/L2 to macrophages was visualized using goat anti-mouse IgG-AF488. Confocal microscopy images are presented as single channels and merges, magnification ×20. Scale bars represent 100 μm. **F, G** E4 attenuates osteoclastogenesis. BMDMs were differentiated for 7 days, followed by 7 days of culture with 50 ng/ml RANKL and 5 μg/ml indicated antibodies before IF staining (n = 5 mice for RANKL group and 3 for others). Osteoclasts were visualized by phalloidin (green) and Hoechst 33342 (blue), confocal microscopy images were presented as merges, scale bars represent 100 μm, and the number of osteoclasts per field was counted. Data are analyzed using one-way ANOVA and presented as mean ± SEM.

outpatient clinic of the Rheumatology Department at the Leiden University Medical Center (LUMC) were included in this study. Patients were selected on high anti-CCP2 titers (>340 U/ml) and none of the patients was treated with B-cell depleting therapies or biologicals. All patients fulfilled the EULAR/ACR 2010 criteria for the classification of RA. The recruitment was approved by the local ethics committee of the LUMC, the Netherlands.

The synovial fluid samples were retrieved from female RA patients fulfilling both the ACR and EULAR classification criteria for ser-opositive RA at clinical visits due to a need for joint effusions, with age range 27–47 years and disease duration 1–16 years, all samples were positive for CEP1[74]. The study was approved by Karolinska University Hospital.

The cartilage explant samples originated from individuals with fractured femur head (HC01) or RA patients (RA01 and RA02), approved by the ethical committee (Dnr 334-15, 2015-05-18; T1075-17, 2017-12-18 2019-04373 2019-09-11)

The human thymic tissues (VSD1 and VSD2) were obtained from two children (64d old female and 153d old male), diagnosed with ventricular septal defect (VSD), and underwent corrective cardiac surgery corrective cardiac surgery at Sahlgrenska University Hospital, Gothenburg, Sweden. Parents gave informed consent, and the study was approved by the Regional Ethical Board at the University of Gothenburg (no. 217-12, 2012-04-26).

All patients gave written informed consent prior to inclusion.

### Animals
The animal models were achieved using various conventional or bred-in-house mouse strains with established technical feasibility (Table S1). In general, male mice older than 8 weeks were used for all experiments unless specified. All mice were kept and bred in climate-controlled, specific pathogen-free (following the Felasa II protocol) environment in the facility Comparative Medicine's Annex (KM-A) of Karolinska Institutet (Stockholm, Sweden) and at the specific pathogen-free animal facility of Medicon Village (Lund, Sweden). The DBA/1 mice used in this study were purchased from Janvier Labs (#SC-DBA1-M). All animals were fed with standard rodent chow and given water ad libitum. Different experimental groups were housed together to minimize experimental bias. All mice were euthanized by $CO_2$. The local ethics committees approved all animal experiments (Stockholm Norra Djur-försöksetiska Nämnd, Stockholm, Sweden and Malmö/Lund Animal Care and Use Committee, Sweden). All in vivo arthritis experiments were covered by the ethical permits N35/16, 2660-2021, 02896-20 and 12369-2018.

### Peptide synthesis and suspension bead array
The library of 17-mer cyclic peptides derived from human COL2 as well as other classic peptides were previously described[11]. Similar design also applies to the included homocitrulline COL2 and CCP4 peptides. Antibody response was analyzed by the multiplex bead-based array (Luminex) by following the previous study[11]. Briefly, MagPlex® beads (Luminex Corporation) with unique IDs were activated by sulfo-NHS (24510, Thermo Fisher Scientific) combining with EDC (22980, Thermo Fisher Scientific), then NeutrAvidin (31000, Thermo Fisher Scientific) was conjugated to the beads, and each biotinylated peptide was cou-pled to designated bead ID (1 peptide/bead ID). The beads were mixed and incubated with indicated antibodies, followed by incubation with 1:750 diluted goat anti-mouse IgG secondary antibody conjugated with R-Phycoerythrin (115-116-071, Jackson ImmunoResearch). The antibody reactivity to the peptides was detected by Bio-plex 200 system (Bio-Rad) and processed by Bio-plex Manager 6.2 (Bio-Rad), the median fluorescence intensity (MFI) values at 1.0 μg/ml antibody concentration were presented.

### Expression and purification of antibodies
Chimeric antibodies (Table 1) containing human variable and mouse constant domain were designed. The mouse IgG2b constant region sequence was obtained from UniProtKB with accession number P01867 and the mouse lambda-1 P01843. First, vectors containing the mouse IgG2b heavy chain (HC) constant region and the lambda light chain (LC) constant region, were created. The variable regions of HC and LC from RA patients were then inserted in the frame before the mouse constant region. Four DNA fragments were synthesized at Eurofins with restriction sites at the 5′ and 3′ ends. (1) The mIgG2b constant region with restriction sites NheI and *BamHI*. (2) The lambda constant region with *HindIII* and *BamHI*. (3) The HC variable region with *KpnI* and *NheI*. And (4) The LC variable region with *KpnI* and *HindIII*. The synthesized genes (constant regions of both HC and LC) were digested using FastDigest™ restriction enzymes (Thermo Fisher Scientific). The digested DNA fragments were cloned into the mammalian expression vector pCEP4 (V04450, Thermo Fisher Scientific) that was digested using the same restriction enzymes. After ligation, two vectors containing mouse IgG2b and lambda constant regions were obtained, designated pCEP4-mIgG2b and pCEP4-mL, respectively. The HC and LC plasmids were co-transfected into Expi293F cells (A14527, Thermo Fisher Scientific)) with FectoPRO™ DNA transfection reagent (Polyplus transfection). The supernatants were harvested 6 days post-transfection. The chimeric antibodies were purified using a 5 mL HiTrap Protein G HP affinity col-umn (GE Healthcare Life Sciences) according to the manufacturer's instruction, or protein G-based affinity chromatography with the ÄKTA™ system. The purified antibodies were buffer-exchanged to PBS solution and the endotoxin was determined using Pierce™LAL Chro-mogenic Endotoxin Quantitation Kit (88282, Thermo Fisher Scientific) with <0.1 EU/mg protein. The antibodies that were traditionally expressed by hybridomas (Table 1) were similarly purified following the hybridoma subcloning and single-clone expansion.

### Antibody-induced arthritis
For the functional study of ACPA in collagen antibody-induced arthritis (CAIA), indicated dose (2, 3 or 4 mg/mouse) of each monoclonal

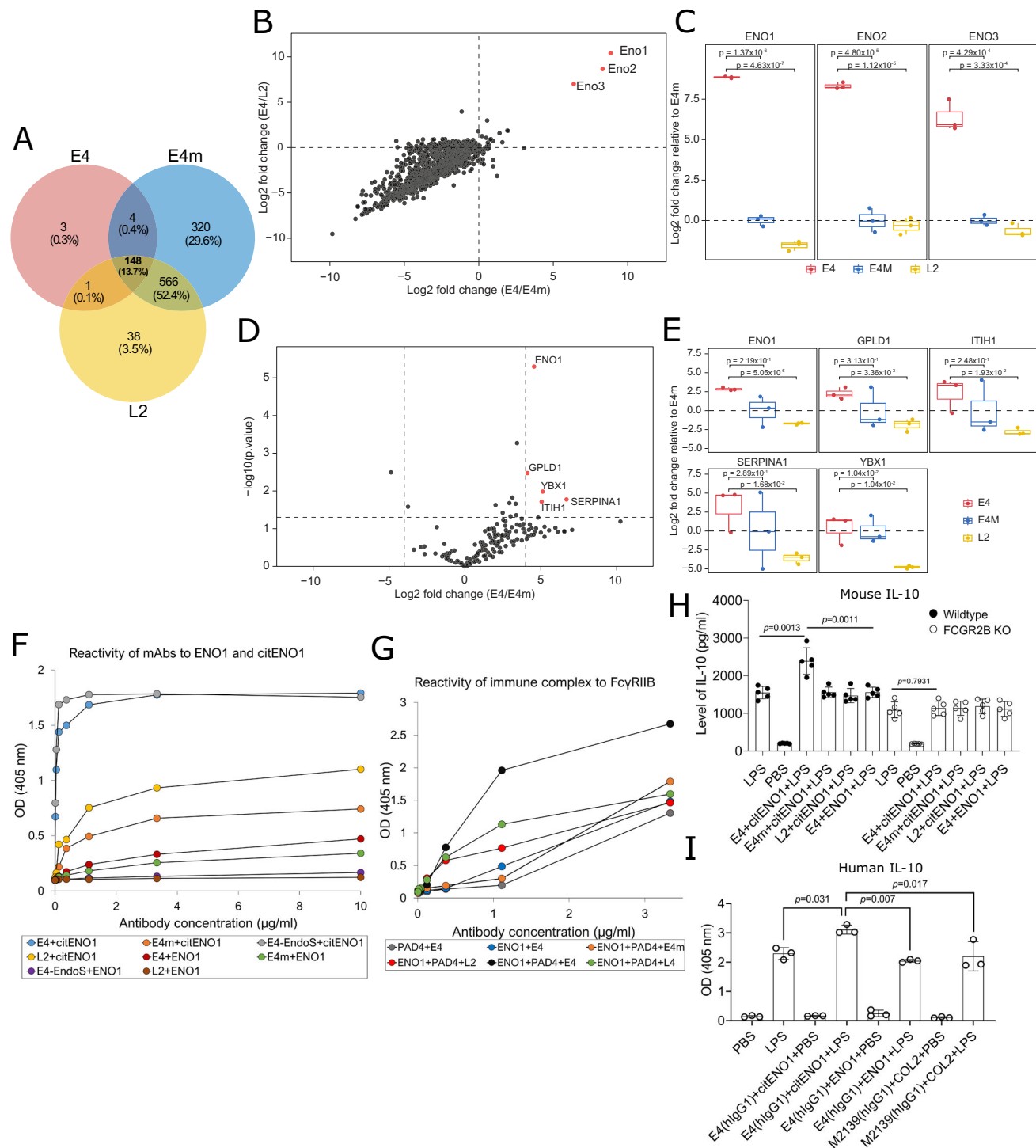

antibody were intravenously or intraperitoneally injected into designated mice on day 0. The mice received a boost by intraperitoneally injecting 25 μg of lipopolysaccharide (LPS, O55:B5) on day 3 or 5. The development of arthritis was monitored in different time points using an extended scoring system. Briefly, each inflamed (swollen or red) toe or knuckle was given 1 point, and each inflamed wrist or ankle was given 5 points, resulting in a maximum score of 15 (5 toes + 5 knuckles+ 1 wrist/ankle) per limb and 60 in total for each mouse.

**Evoked pain-like behavior test**
For the result shown in Fig. 2B, a dose of 4 mg of ACC4 and 2 mg of ACC1 or E4 were intravenously injected into male B10.RIII (ACC4 and saline) or female Balb/c (ACC1 and E4) mice on day 0 (n = 6 or 12 per

group), whereas the control animals received equal volume of saline. Mechanical hypersensitivity in the hind paws was used as measure of evoked pain-like behavior and assessed using von Frey filaments. Mice were habituated to the testing environment, individual compartments on top of a wire-mesh surface (Ugo Basile), prior to baseline testing. Three baselines were recorded and averaged prior to injection of antibodies. On the test days, mice were allowed to acclimatize to the testing cages for 1 h prior to testing. Withdrawal thresholds were measured with von Frey OptiHair filaments (Marstock OptiHair) of logarithmic increasing stiffness (0.5, 1, 2, 4, 8, 16, and 32 mN, corresponding to 0.051, 0.102, 0.204, 0.408, 0.815, 1.63 and 3.26 g, respectively). Filaments were pressed perpendicularly against the plantar surface of the hind paw and a positive response was considered

**Fig. 6 | Reactivity of E4 ACPA to macrophage and RA synovial fluid (SF) proteins. A**–**C** Reactivity of E4 to mouse macrophage proteins. Naïve bone marrow-derived macrophages (BMDMs) ($n = 3$) were stimulated with LPS (500 ng/ml) overnight, and proteins were extracted for immunoprecipitation (IP) with ACPA. Numbers of captured proteins identified by mass spectrometry from each antibody are presented as (**A**) Venn diagram; correlation between log2-fold-change of E4/E4m (x-axis) and E4/L2 (y-axis) in (**B**) scatter plot; and proteins of interest in (**C**) boxplots. **D**, **E** Reactivity of E4 to human SF proteins. The CEP1-positive RA SF samples were subjected to IP with E4, L2 or E4m ($n = 3$). Captured proteins were presented as log2-fold-change (x-axis) and −Log10 of p value ($p < 0.05$, y-axis) in (**D**) volcano plot; and proteins of interest in (**E**) boxplots. Horizonal lines in the boxplots represent the median, 25th and 75th percentiles and whiskers represent measurements to the 5th and 95th percentiles. **F** Reactivity of ACPA to citrullinated human α-enolase (citENO1). ENO1 was citrullinated by hPAD4 with 100 mM $Ca^{2+}$ and coated on ELISA plates. Antibody binding was detected using anti-mouse IgG Fc-

HRP antibody with ABTS assay, results were presented as curve plot. **G** Reactivity of ACPAs to FCGR2B. CitENO1 was incubated with indicated biotinylated antibodies to form immune complexes (ICs). ICs were added to ELISA plate coated with recombinant FCGR2B and detected by Streptavidin-HRP. For unmodified ENO1/PAD4 controls, either PAD4 without ENO1 or ENO1 without PAD4 were used. Results were presented as curve plot. **H** E4-citENO1 IC increases IL-10 secretion from mouse macrophages. Wildtype or FCGR2B KO macrophages ($n = 5$) were treated with similarly prepared ICs (10 μg/ml) or LPS (500 ng/ml) overnight. Supernatants were concentrated for ELISA using the Europium assay. **I** E4 human IgG1 in complex with citENO1 increases IL-10 secretion from human macrophages. The hIgG1 E4-citENO1 IC was prepared similarly, the hIgG1 M2139-COL2 IC was used as control. The human CD14⁺CD16⁻ monocyte-derived macrophages were treated with IC (10 μg/ml) or LPS (250 ng/ml) overnight ($n = 3$) and supernatants were collected for ELISA developed by ABTS assay. ELISA data are analyzed using one-way ANOVA and presented as mean ± SD.

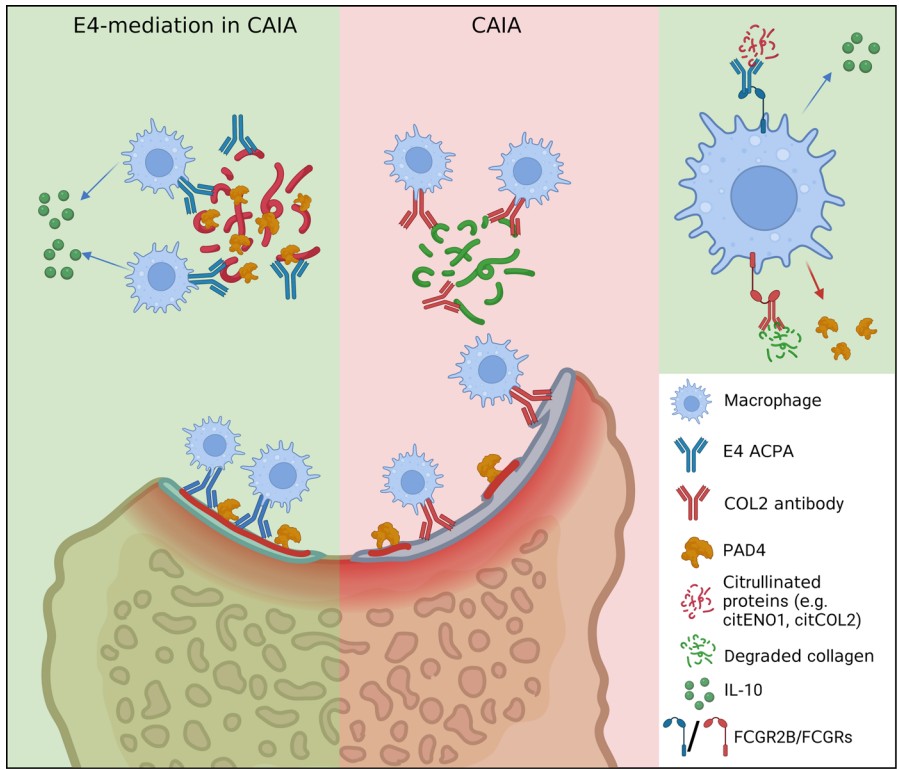

**Fig. 7 | Schematic proposal for the function of E4 ACPA in effector arthritis.** In collagen antibody-induced arthritis (CAIA), macrophages are activated by LPS-mediated TLR signaling and arthritogenic immune complexes, leading to an increased secretion of PAD4. PAD4-mediated citrullination generates antigens that could be recognized by ACPAs. E4 binds to certain group of citrullinated antigens

(e.g., ENO1) and forms local immune complexes that preferably interact with the FCGR2B on activated macrophages, delivering regulatory signals to macrophages, promoting IL-10 secretion and reducing osteoclastogenesis. The graph was created and licensed with Biorender.com.

---

if a brisk withdrawal of the paw from the filament was noted within 2–3 seconds of application. The 50% withdrawal threshold (force of the filaments necessary to produce a reaction from the animal in 50% of the applications) was calculated using the Dixon up–down method and the results were expressed in grams[30]. Withdrawal thresholds from both hind paws were determined and averaged. Assessment of mechanical hypersensitivity was performed between 10:00 – 17:00.

For the results shown in Supplementary Fig. 2, a dose of 4 mg of ACC1 or E4 were intravenously injected into C57BL/6 N mice on day 0 ($n = 5$ or 6 per group), whereas the control animals received 150 μl of saline. On day 3, mechanical allodynia was assessed by the paw withdrawal response to von Frey Filament using the up–down method. A series of filaments with a logarithmically incremental increasing stiffness of 0.04, 0.07, 0.16, 0.40, 0.60, 1.0, and 2.0 (g) were applied to the

plantar surface of the hind-paws and held for 2–3 s. A 2.0 g-filament cutoff was used to avoid tissue damage and brisk paw withdrawal was considered a positive response. The 50% probability withdrawal threshold was calculated.

The investigators were blinded to the origin and treatment of the mice throughout behavioral assessments and data analysis in both experiments.

## G6PI-induced arthritis

The glucose-6-phosphate isomerase (G6PI) induced arthritis (GPIA) was achieved by immunizing the wildtype DBA/1 mice with hGPI325-339 peptide. On day 0, prior to peptide immunization, E4 antibody (3 mg/mouse, $n = 9$) or equal volume of PBS ($n = 15$) were intravenously injected into the mice, then 0.2 mg/ml of peptide was 1:1

mixed with CFA, 100 µl of the mixture was subcutaneously injected to each mouse at the base of tail (10 µg/mouse). The arthritis severity was monitored for 28 days using the extended scoring system as described in CAIA experiments.

## Mouse carrageenan air-pouch model

To establish air-pouch in mice, on day 0, 5 ml of clean air were subcutaneously injected into the shaved back of the animals together with intraperitoneal injection of 2 mg E4 or equal volume of PBS. After 3 days, another 2 ml of clean air were similarly injected into each mouse for compensation. On day 6, 1 ml of 2% λ-carrageenan (22049, Sigma-Aldrich) was injected into the pouch of each mouse. The next day, immediately after the mice were sacrificed, the air-pouch was sagittally cut open and 1.5 ml of PBS with 5.5 mM EDTA were applied to wash the pouch interior, the lavage fluid was collected, and volumes recorded. After counting the cell number, samples were centrifuged at 1000 ×g for 10 min, cell pellets were resuspended by PBS and subjected to flow cytometry analysis.

## Experimental autoimmune encephalomyelitis (EAE) in mice

On day 0, the $MOG_{35-55}$ peptide (3038001, MD Bioproducts) was dissolved in PBS to 4 mg/ml. CFA is prepared by mixing H37Ra (231141, BD Biosciences) in IFA (263910, BD Biosciences) to a concentration of 6 mg/ml. The MOG-CFA emulsion was prepared by 1:1 mixed MOG and CFA, quality-checked by drop test then subcutaneously injected in a total volume of 100 µl to each mouse. After 2 h, 200 ng of Pertussis toxin (PTX) diluted in 200 µl of PBS were intraperitoneally injected into each mouse as a booster. On day 1, 1 mg of E4 antibody (n = 9) or equal volume of PBS (n = 10) was intravenously injected into the mice. Mice received another boost of PTX on day 2 and the same dose of E4 antibody on day 9. Mice were weighed and the disease was macroscopically scored in different time points, briefly, animals with EAE develop ascending flaccid paralysis that initially affects the tail (score 1–2), later involves hind limbs (score 3-6), forelimbs (score 7) and ultimately result in quadriplegia and death (score 8).

## Mannan-induced psoriasis (MIP)

Psoriasis was induced by intraperitoneal injection of 20 mg of mannan (M87504, Sigma-Aldrich) for each mouse, the antibody treatment group (n = 5) at the same day received 3 mg of E4 antibody intravenously. Clinical disease was monitored daily until day 9. The skin inflammation was evaluated by scoring scaling of the skin of the hind paws using a macroscopic scoring system ranging from 0 to 3 (0=no scaling, 1=mild scaling, 2= medium scaling, 3= severe scaling), and the redness was similarly evaluated.

## Surface plasmon resonance (SPR) analysis

The binding of E4 to the citrullinated histone H2A was tested by SPR using Biacore 2000 (GE Health). In short, the biotinylated peptide was coupled to an SM sensor chip by streptavidin binding, the full-length E4 and control antibody F3 were diluted in PBS-P running buffer (20 mM phosphate buffer, 2.7 mM KCl, 137 mM NaCl and 0.05 % Surfactant P20) and injected at 30 µL/min in a series of concentrations. Data were processed using Biacore 2000 evaluation software.

## Biotinylation and EndoS treatment of antibodies

The EZ-Link NHS-LC-Biotin (Thermo Fisher Scientific) was used for biotinylation of indicated antibodies, the reaction was carried out by incubating 20-fold molar excess of biotin reagent with the antibody at room temperature for 30 min. For the Fc N-glycan cleavage, GST-fused endoglycosidase S (EndoS) expressed by *E.coli* was used to incubate with E4 antibody at a ratio of 1:1000 (w/w) and 37 °C for 1 hour. All antibodies were purified by using Protein G GraviTrap Columns (VWR) according to the manufacturer's instructions.

## ATDC5 cell culture

The chondrogenic cells ATDC5 (Merck) derived from mouse teratocarcinoma AT805 were seeded in 96-well black µ-plate ($2.5 \times 10^4$ cells/well) and cultured for ≥14 days in the medium containing a 1:1 mix of DMEM GlutaMAX™ (Gibco) and Ham's F-12 Nutrient Mix GlutaMAX™(Gibco) supplemented with 5% FBS, 1% penicillin-streptomycin and 1× insulin-transferrin-selenium (ITS-G, Gibco). The culture medium was changed every 2–3 days. On day 14, 1) for the antibody binding assay, 10 µg/ml of recombinant human PAD4 (Protein Science Facility, Karolinska Institutet) or PBS were added to the culture for citrullination overnight with the supplement of 10 mM $Ca^{2+}$, after that, indicated antibodies (10 µg/ml) were added for another hour of incubation before IF assay.

## Histology

To determine the mAb reactivity with joint tissue in vivo, 2-day-old neonatal mice were intraperitoneally injected with 100 µg of biotinylated mAbs. We used neonatal mice as they lack bone in the joints and are therefore easier for sectioning and a decalcification step is normally not needed, thus they are more suitable to observe the in vivo binding of injected antibodies[75]. After 48 h, the knee joints were snap-frozen in isopentane on dry ice and stored at −80 °C. Joint sections (5 um) were fixed in 4% paraformaldehyde for 5 min, rinsed in PBS, blocked for endogenous peroxidase for 30 min (0.5% $H_2O_2$ with 0.1% Tween 20), incubated with Extravidin peroxidase (E2886, Sigma-Aldrich) for 30 min, and developed with diaminobenzidine (DAB kit (Dako Omnis), Agilent) for 8–9 min.

To assess the binding of mAbs to the paraffin-embedded joint sections in vitro, knee joints and paws from adult naïve or CAIA mice were dissected, decalcified, dehydrated and paraffin-embedded. Sections of a thickness of 5 um were stained with hematoxylin/eosin/toluidine blue. For immunohistochemical staining, in general, sections were deparaffinized in xylene, hydrated in 100% then 95% EtOH, rinsed in distilled water, incubated with HistoReveal (ab103720, Abcam) for antigen retrieval, blocked by 1% BSA-PBS and goat anti-mouse IgG (H + L) (1:10) (115-007-003, Jackson ImmunoResearch) at 4 °C overnight, incubated with given mAbs (20 µg/ml) at 4 °C overnight then secondary antibody (goat anti-mouse IgG(H + L)-CF™488 A antibody (SAB4600388, Sigma-Aldrich) or goat-anti-mouse IgG1-Alexa Fluor™ 488 antibody (A21121, Thermo Fisher Scientific) at 1:1000 and room temperature for 1 hour, dehydrated in 95% then 100% EtOH, and mounted to slides with ProLong™ Glass Antifade Mountant with NucBlue™ Stain (P36983, Thermo Fisher Scientific) before microscopy observation. Sections were washed 3 times by PBS-T (0.05% Tween 20) between steps. The human cartilage explants were cryosectioned to 7 um and fixed in ice-cold acetone for 5 min, followed by air-drying for 30 min. Samples were blocked by 1% BSA-PBS with 0.5% Triton™ X-100 (X100, Sigma-Aldrich) for 1 hour at room temperature, then incubated with given antibodies (20 µg/ml) at 4 °C overnight, and detected with secondary antibody (goat anti-mouse IgG(H + L)-CF™488 A antibody (SAB4600388, Sigma-Aldrich) at 1:1000. The slides were mounted with ProLong™ Glass Antifade Mountant with NucBlue™ Stain (P36983, Thermo Fisher Scientific) formulated with Hoechst 33342.

Human and murine thymus tissue embedded in OCT compound was cut into 7 µm sections with a cryostat. The sections were fixed with cold acetone and blocked with Protein Block (X0909, Dako). The human and the naïve murine samples were stained at 4 °C for 1 h with biotinylated ACC1, ACC4, E4NG, and M2139. The tissues were washed and incubated at 4 °C for 30 min with Streptavidin Alexa Fluor 555 (S21381, Thermo Fisher Scientific) at 1:200 and Hoechst (H21486, Thermo Fisher Scientific) at 1:5000. The pre-injected murine samples with given antibodies were only stained with streptavidin and Hoechst. Sections were mounted with ProLong Gold Antifade Mountant (P36930, Thermo Fisher Scientific).

For ATDC5 cell culture staining, after incubation with PAD4, cells were fixed in 2% PFA at room temperature for 10 min. Indicated antibodies (E4, E4m or L2) (20 µg/ml) were incubated with the fixed cells for 1 hour, followed by blocking step with 1% BSA buffer. To detect E4, E4m and L2, anti-mouse IgG secondary antibody (SAB4600388, Sigma Aldrich) at 1:1000 was used. To detect the citrullinated COL2 α chain in the extracellular matrix, biotinylated ACC4 antibody was used and detected by streptavidin-PE at 1:500. Hoechst33342 at 1:5000 was used for nuclei detection. For macrophage staining, cells were fixed and incubated with indicated antibodies as above. The 2.4G2 antibody (553141, BD Biosciences) was used at 1:200 for blocking the Fc receptors. Antibodies were detected by anti-mouse IgG(H + L)-CF™488 A antibody (SAB4600388, Sigma-Aldrich) at 1:1000, macrophages were stained with F4/80 antibody conjugated with APC (123116, Biolegend) at 1:200 and DNA was stained with Hoechst33342 at 1:5000. For osteoclast staining, phalloidin-iFluor 488 reagent (ab176753, Abcam) at 1:1000 was used for detecting actin filaments and nuclei were stained with Hoechst33342 at 1:5000.

All cells for immunofluorescence imaging were cultured in the black 96-well µ-Plate (89626, Ibidi), samples were washed for at least three times between each step, immunofluorescence imaging was performed using LSM700/800 confocal microscope (Carl Zeiss AG) and images were processed by ZEN 3.0 (Carl Zeiss AG) and ImageJ 1.52 h software.

### Citrullination in vitro and generation of immune complexes

Substrates such as recombinant human α-enolase (ENO1) or bovine collagen type II (COL2) were citrullinated by recombinant human PADI4 (Protein Science Facility, Karolinska Institutet) at an E/S ratio 1:20 (ENO1) or 1:10 (COL2) in 0.1 M Tris-HCl buffer (pH = 7.4) containing 0.1 M $Ca^{2+}$, the incubation took overnight at 37 °C. To generate the ACPA-citENO1 immune complexes, antibodies with or without biotinylation (depending on the assay) were added to above-mentioned mixture after citrullination (Ab/Ag molar ratio = 2:1), and allowed to incubate for 2 h at RT before further uses.

### Differentiation of murine bone marrow-derived macrophages (BMDMs) and osteoclasts

For in vitro differentiation of BMDMs, depending on the purpose, bone marrow cells were obtained from Balb/c, B10.Q or FCGR2B knockout mice, and $(2-3) \times 10^6$ cells/well were seeded in 6-well culture plates (83.3920, Sarstedt) for protein extraction, $10^6$ cells/well in 24-well plates for cytokine detection (83.3922, Sarstedt), or $10^4$ cells/well in 96-well black µ-plate (ibidi) for IF assay. The RPMI 1640 GlutaMAX™ medium (Gibco) containing 10% FBS, 1% penicillin-streptomycin and 20 ng/ml of recombinant murine M-CSF (Peprotech) was used for the differentiation, the medium was changed every 2–3 days, cells were cultured for ≥5 days at 37 °C in a conventional $CO_2$ incubator until adherent BMDMs were fully differentiated. BMDMs were treated differently depending on the assay. For protein extraction, BMDMs were stimulated overnight with LPS (500 ng/ml) or PBS. For cytokine detection, cells were treated with indicated immune complexes/antibodies treatment (10 µg/ml) and LPS (500 ng/ml) overnight, the next day, supernatants were collected and concentrated using Vivaspin 6, 10 kDa MWCO columns (28-9322-96, GE Healthcare) before sandwich ELISA. For IF assays, BMDMs were treated with indicated antibodies (20 µg/ml) and LPS (100 ng/ml) overnight beforehand. For the differentiation of osteoclasts, BMDMs were similarly differentiated, on day 7, 50 ng/ml of recombinant murine RANKL (Peprotech) combined with 5 µg/ml of indicated antibodies were added for another 7 days of culture until IF staining on day 14.

### Isolation of human monocytes and differentiation of human macrophages

Buffy coats from healthy donors were obtained from Karolinska University Hospital (Solna, Stockholm), the peripheral blood mononuclear cells (PBMCs) were isolated via gradient centrifugation using Ficoll-Paque Plus (17144002, Cytiva) in SepMate™-50 tubes (85450, Stemcell Technologies). The CD14+CD16- monocytes were isolated using the Human Classical Monocyte Isolation Kit (130-117-337, Miltenyi Biotec) by following the kit instructions. The magnetically labeled non-target cells were depleted using a MACS® column while the unlabeled CD14+CD16- monocytes ran through the column and were collected. To differentiate human macrophages from the isolated monocytes, $0.5 \times 10^6$ monocytes/well were seeded in 24-well plate with the culture medium RPMI 1640 GlutaMAX™ medium (Gibco) containing 10% FBS, 1% penicillin-streptomycin and 40 ng/ml of recombinant human M-CSF (PHC9504, Thermo Fisher Scientific). Cells were cultured for 10 days at 37 °C in conventional $CO_2$ incubator with changes of fresh culture medium every 3–4 days. On day 10, differentiated macrophages were stimulated by LPS (250 ng/ml) or PBS together with indicated hIgG1 E4 or M2139 immune complexes (10 µg/ml) overnight. The supernatant was then collected for cytokine detection.

### ELISA

An in-house method was developed to identify ACPA-positivity in the single B cell cultures. In brief, 1 µg/ml biotinylated-CCP2 and 10 µg/ml biotinylated-CArgP2 were coupled to streptavidin-coated Nunc Maxisorp plates (Fisher, #430341) via incubation at RT for 1 h. Culture supernatants were tested in a 1:2 dilution with PBS/1%BSA/0.05% tween (PBT) and samples were incubated for 1 h at 37 °C. ACPA-IgG binding was detected using an HRP-conjugated rabbit-anti-human IgG secondary antibody (DAKO, P0214). ELISA read-out was performed using ABTS and $H_2O_2$. Reactivity to CCP2 was determined based on the arginine version of the peptide. For sandwich ELISA detecting mouse/human cytokines, 5 µg/ml of IL-10 antibody (551215, BD Biosciences for mouse or 506802, Biolegend for human) or TNF antibody (559064, BD Biosciences for mouse) were coated in Nunc Maxisorp plates for overnight at 4 °C. The plates were blocked by 5% non-fat milk for 1 hour at RT, the concentrated supernatants from macrophage cultures were then added to the plates and incubated for 2 h at RT, followed by another same incubation with 2 µg/ml biotinylated IL-10 antibody (554423, BD Biosciences for mouse or 501502, Biolegend for human) or TNF antibody (558415, BD Biosciences for mouse). The results were developed using 1:1000 diluted Europium-labeled streptavidin (1244-360, PerkinElmer) or 1:2000 diluted Streptavidin-HRP (554066, BD Biosciences) in an ABTS assay. For ELISA detecting antibody-antigen binding, rabbit-anti-human ENO1 antibody (PA130493, Thermo Fisher Scientific) was coated in Nunc Maxisorp plates (5 µg/ml) overnight at 4 °C. The next day, plate was blocked with 5% non-fat milk and recombinant human ENO1 citrullinated by human PAD4 was added for 2 h of incubation at room temperature, followed by the incubation with indicated antibodies with different concentrations for 2 h at RT, antibodies were detected using goat anti-mouse IgG (H + L) secondary antibody conjugated with HRP (115-035-003, Jackson ImmunoResearch) at 1:5000 by ABTS assay. For detecting antibody binding to COL2, citrullinated bovine COL2 (5 µg/ml) was directly coated to the plate, the following steps were similar as above. For detecting IC-FCGR2B binding, the plates were similarly coated with 5 µg/ml recombinant FCGR2B (produced in-house) and blocked. The generated immune complexes (biotin-ACPA-citENO1) or various control conditions (without PAD4 or ENO1) in different concentrations were applied, incubated for 2 h at RT, and detected using streptavidin-HRP (554066, BD Biosciences) at 1:2000 by ABTS assay.

### Flow cytometry

Bone marrow-derived macrophages (BMDMs) were differentiated for 5 days in 6-well plates followed by the treatment of LPS (100 ng/ml) or PBS overnight as abovementioned. The next day, 20 µg/ml of indicated biotinylated antibodies were added to the culture and incubated for

60 min. Cells were washed thoroughly and transferred to 96-well plate. Cells were stained using the Live/Dead™ Fixable Near-IR Dead Cell Stain Kit (Thermo Fisher Scientific) at 1:1000 for 30 min at room temperature, blocked by anti-CD16/32 antibody (clone: 2.4G2, BD Biosciences) at 1:100 for 5–10 min on ice in the dark, and followed by staining with antibodies at 1:200, including CD38-FITC (102705, Biolegend), CD11b-PE (557397, BD Biosciences) and F4/80-PerCP-Cy5.5 (45-4801-82, Thermo Fisher Scientific) for 30 min at room temperature. Cells were then stained with streptavidin-APC (554067, BD Biosciences) at 1:500 for 15 min, and fixed by Cytofix/Cytoperm™ (554722, BD Biosciences) on ice for 20 min, according to the kit instructions. For cells from mouse air-pouch lavage fluid, CD11b-FITC (553310, BD Biosciences), F4/80-PerCP-Cy5.5 (45-4801-82, Thermo Fisher Scientific), Ly6C-APC (128016, Biolegend) and Ly6G-PE (127608, Biolegend) antibodies at 1:200 were used. For the ex vivo staining of splenocytes, spleens from naïve and CAIA mice injected with given biotinylated antibodies were obtained and RBCs were processed by ACK lysing buffer before staining. CD11b-PB(101224, Biolegend), CD11c-PE-Cy7 (558079, BD Biosciences), Ly6G-PE (127608, Biolegend), Ly6C-BV605 (128035, Biolegend), F4/80-PerCP-Cy5.5 (45-4801-82, Thermo Fisher Scientific), MHCII-FITC(553623, BD Biosciences), CD45-AF700 (103128, Biolegend) at 1:200 and streptavidin-APC (554067, BD Biosciences) at 1:500. Flow cytometry was carried out using LSRII (BD Biosciences) or Attune (Thermo Fisher Scientific) cytometer and data were analyzed using FlowJo software.

### Macrophage protein extraction and synovial fluid processing

Macrophages differentiated in 6-well plates were stimulated by 500 ng/ml of LPS overnight. Cells were thoroughly washed and harvested using StemPro™ Accutase™ (Thermo Fisher Scientific), proteins from the cells (membrane/cytosolic) were extracted using Mem-PER™ Plus Membrane Protein Extraction Kit (Thermo Fisher Scientific) according to kit instructions. For human synovial fluid, samples were processed by ~1500 U/ml of hyaluronidase from bovine testes (Sigma Aldrich) at 37 °C for 15 min, samples were then centrifuged at 1000 × g and room temperature for 10 min, the supernatant was collected and 1:1 diluted by PBS. All samples were stored at −80 °C.

### Immunoprecipitation

To precipitate the targets of given antibodies, 10 µg of each indicated biotinylated antibodies were immobilized to streptavidin-coated magnetic beads (Dynabeads™ MyOne™ Streptavidin C1/T1, Thermo Fisher Scientific) according to the manufacturer's instructions. Extracted proteins from macrophages or human synovial fluid were subjected to incubation with the beads for 2 h at room temperature with gentle rotation. The beads were washed between each step 3–4 times by sterile PBS. The proteins that were captured by the beads were then eluted and stored at −80 °C before mass spec analysis.

### Sample preparation for proteomics analysis

Samples were resuspended in 50 mM Tris at pH 8.5 and 2 M urea and protein concentration was determined by BCA (Thermo Fisher Scientific) according to the manufacturer's instructions. Samples were incubated for 1 h at room temperature with 5 mM Dithiothreitol, followed by 15 mM of iodoacetamide for 1 h in the dark. For Macrophage samples, urea concentration was reduced to 1 M and 1:50 (w/w) Trypsin (Promega) was added to each sample for overnight digestion. Human synovial fluid samples were digested for 2 h with 1:100 (w/w) LysC (Fujifilm Wako), then urea concentration was reduced to 1 M and 1:100 (w/w) Trypsin (Promega) was added to each sample for overnight digestion. Samples were acidified by formic acid until the pH was below 3 and desalted by Pierce™ C18 Tips (Thermo Fischer Scientific).

### Mass spectrometry analysis

Samples were resuspended in Buffer A (2% ACN, 0.1% FA in water) and injected into an UltiMate 3000 UPLC autosampler (Thermo Fisher Scientific) coupled to an Orbitrap Fusion Lumos Tribrid mass spectrometer (Thermo Fisher Scientific). The peptides were loaded on a trap column (Acclaim PepMap 100 C18, 100 µm × 2 cm) and separated on a 50 cm long C18 Easy spray column (Thermo Fisher Scientific). Peptides were separated by an 80 min gradient, 4–26% Buffer B (98% ACN, 0.1% FA in water) in 55 min, followed by an increase to 40 % Buffer B in 10 min, followed by 3 min to 95% Buffer B which was kept for another 3 min and 15 min equilibration phase with a flow rate of 300 nl/min. The parameters are specified in Table S2.

### Bioinformatics

Raw files were processed with MaxQuant (version 1.6.2.3)[76]. For peptides search, acetylation of N-terminal and oxidation of methionine were selected as variable modifications whereas carbamidomethylation of the cysteine was selected as a fixed modification. Trypsin with up to 2 missed cleavages was set as protease and the spectrum was searched against the UniProt *mus musculus* database or the Swissprot *Homo sapiens* database, for the macrophage and human synovium fluids samples, respectively. The FDR was set to 0.01 for both peptides and proteins. The match between run feature and LFQ were enabled. For all other parameters, the default settings were used.

All data analysis and visualization were produced using R version 4.1.1. Contaminants, decoy proteins, and protein only identified by site as well as proteins with less than 2 identified peptides were removed. All proteins with at least three valid LFQ values for one sample group were used for further analysis. Total signal intensity was normalized by median centering and missing values were imputed based on. Missing values were replaced by sampling random numbers from a normal distribution: 1.8 standard deviation down shifted from the mean with a width of 0.25. To calculate p-values, a two-sided $t$ test with equal variance was applied. Proteins with a $p$-value < 0.05 and an absolute log2-scaled foldchange of 4 were considered enriched.

### Statistics and reproducibility

Statistical analysis is specified in each section and in figure legends. R and Prism 9.4.0 (Graphpad Software, Inc.) were used for statistical analysis and plotting. Two-tailed Mann–Whitney or ANOVA test was used for comparison. A $p$-value lower than 0.05 was considered as threshold for statistical significance among control and experimental groups. All replicates in this study are biologically independent samples/animals/cells etc., and experiments were independently repeated at least twice with similar results.

### Reporting summary

Further information on research design is available in the Nature Portfolio Reporting Summary linked to this article.

## Data availability

The mass spectrometry-based proteomics data have been deposited to ProteomeXchange[77] Consortium via the PRIDE partner repository with data set identifier PXD035177 and PXD035180 for the macrophage and human synovium fluids samples, respectively. All other data are included in the Supplemental Information or available from the authors upon reasonable requests, as are unique reagents used in this Article. The raw numbers for charts and graphs are available in the Source Data file whenever possible. Source data are provided with this paper.

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

## Acknowledgements

The authors thank the staffs from the Histocore and Biomedicum Imaging Core (BIC) of Karolinska Institute (KI) for the assistance in histology and imaging, Comparative Medicine's Annex (KM-A) of KI and Animal Facility in Lund for taking care of the mice, Protein Science Facility (PSF) of KI for protein synthesis, Redoxis AB and Zhongwei Xu for help in experiments. Especially, we also want to thank all the donors for the human samples used in this study. The work was supported by grants from The Knut and Alice Wallenberg Foundation (2019-0059), the Swedish Research Council (2019-01209), the China Scholarship Council (CSC), and the Natural Science Foundation of China No.32070913.

## Author contributions

R.H., Y.H., and C.G. conceived and designed the experiments; Y.H., C.G., A.M.G., B.X., C.M.B., K.S., J.S., L.C., E.L., C.L., L.M.S., D.T., V.U., B.L., T.L., G.F.L., M.A., P.E., C.S., and R.A.Z. were involved in conducting experiments, data acquisition, analysis, and interpretation; R.E.M.T, H.U.S, T.R., and L.M.S were involved in isolating monoclonal antibodies from RA patients; V.M provided the human synovial fluid samples; I.G provided human cartilage explants. O.E provided human thymus samples. Y.H

and R.H drafted the manuscript. All authors participated in revising the manuscript and approved the final submission.

## Funding

## Competing interests

C.G., B.X., T.R., and R.H. are the co-inventors of a patent application (PCT/EP2018/082236) related to ACPA specificity and their use in the diagnosis, prevention, and treatment of autoimmune diseases. H.U.S. and R.E.M.T. are mentioned as inventors on a patent on ACPA-IgG V-domain glycosylation. The remaining authors declare no competing interests.

## Additional information

[1]Section for Medical Inflammation Research, Department of Medical Biochemistry and Biophysics, Karolinska Institutet, 171 77 Stockholm, Sweden. [2]Redoxis AB, 223 81 Lund, Sweden. [3]Division of Physiological Chemistry I, Department of Medical Biochemistry and Biophysics, Karolinska Institutet, Stockholm, Sweden. [4]Department of Physiology and Pharmacology, Center for Molecular Medicine, Karolinska Institutet, 17177 Stockholm, Sweden. [5]Division of Molecular Neurobiology, Department of Medical Biochemistry and Biophysics, Karolinska Institutet, 171 77 Stockholm, Sweden. [6]Department of Rheumatology and Inflammation Research, Institute of Medicine, Sahlgrenska Academy, University of Göteborg, Göteborg, Sweden. [7]Department of Rheumatology C1-R, Leiden University Medical Center, PO Box 9600 Leiden 2300RC, The Netherlands. [8]Center for Medical Immunopharmacology Research, Southern Medical University, Guangzhou, China. [9]Department of Medicine, Division of Rheumatology, Center for Molecular Medicine, Karolinska Institutet, Karolinska University Hospital, Stockholm, Sweden. [10]Department of Immunopathology, Sanquin Research and Landsteiner Laboratory, Academic Medical Center, University of Amsterdam, 1066 CX Amsterdam, the Netherlands. [11]Department of Pediatrics, Institute of Clinical Sciences, Sahlgrenska Academy, University of Göteborg, Göteborg, Sweden. ✉e-mail: Rikard.Holmdahl@ki.se

