## [Peer Review File · Nature Communications]

A subset of antibodies targeting citrullinated proteins confers protection from rheumatoid arthritisREVIEWER COMMENTS

Reviewer #1 (expertise in inflammation and macrophages in rheumatoid arthritis):

In this manuscript a number of chimeric antibodies or mutants were developed from the PB of patients with ACPA+ RA, which contain a human variable domain and a mouse IgG2b constant domain. Also employed were mouse hybridomas that bound citCol2 (IgG2b or IgG1) and a murine arthritogenic antibody to Col2. One of these chimeric antibodies, E4, was observed to suppress CAIA, bind also to ENO1-3 and CitENO1. E4-CitENO1 complexes were able to induce IL-10 more avidly through FCGR2B, and when injected into mice were able to suppress CAIA and GPI induced arthritis, but not other forms of inflammation. E4 was able to bind LPS stimulated macrophages to the same degree as L2, only E4 was effective at decreasing RANKL induced osteoclastogenesis in vitro. L2 appeared less effective than E4 at binding RA cartilage and CAIA cartilage, even though it had binds to more citrullinated epitopes.

The observation that ACPA may reduce inflammation is novel and potentially important. However, there are a number of concerns with the manuscript as presented.

1) the notion that there is no data supporting the concept that ACPA may contribute to disease pathogenesis is not correct. There is substantial evidence that ACPA may contribute to disease pathogenesis, however its role is not conclusive.

2) The conclusions are based on only 1 chimeric antibody. It is difficult to generalize based on an n =1.

3) The effects of this antibody are based on a mouse IgG constant region, which is necessary for the experiments described to be performed. However, since Fc glycosylation and sialylation are important in the pathogenic potential of ACPA, it is a concern that some confirmatory experiments using an antibody with a human Fc component were not performed.

4) Does E4 interfere with the binding of M2139 to cartilage?

5) In panel A of Figure 1 the identification of what the symbols represent is unclear, primarily because of the placement of LPS.

6) Figure 1 E should include the quantification of the histologic abnormalities: inflammation etc.

7) Figures 3 A-B – clarify that mice were induced with CAIA or not and then the cartilage stained with E4 etc. Also, if the cartilage was stained after CAIA from mice not treated, why would the cartilage from the FCGR2B deficient mice not stain with E4?

8) Is Figure 3C is not clear to me. What is represented by ACPA? If this is supposed to show co-expression, I could not see it.

9) The inhibition of CAIA seems much less in Figure 5A compared to earlier figures. How variable is the suppression.

10) In Figure 5E, I do not appreciate the co-expression of F4/80 and ACPA, if that is the purpose.

11) Figure 4A and C, might be more clear if a more high power view of the relevant data were included.

12) Figure 6 C and E, are very difficult for me to read even when magnified. The color combination does not show well, but as best I can see the data, it seems to be repetitious of panel B.

13) Figure 6F is not clear to me. The title and legends suggests that the read out is antibodies binding to cit-ENO1? Maybe the title to the panel should read binding to ENO1 or citENO1?

14) Figure S5. The point is not clear nor is the data convincing. Is the binding thought to be through Fc or Fab binding. In panel A, I don't see co-expression and panel B the staining for CD11c and CD11b should demonstrate a negative population.

15) Figure S7: the macrophages may already have been maximally stimulated to produce TNFa. Do the complexes induce TNFa in macrophages not stimulated with LPS or with a suboptimal dose. A positive control to show that it is possible to induce more TNF by other complexes should be included.

16) Why E4 and not other antibodies with similar specificities and Fc domains did not demonstrate the reported effect should be more convincingly discussed.

Reviewer #3 (expertise in rheumatoid arthritis and ACPA):

Anti-Citrullinated Protein Antibodies (ACPA) is a hallmark of RA. The authors expressed monoclonal ACPAs from RA patients, in mice. Antibodies did not induce pain or RA. One antibody, E4,

protected against antibody-induced arthritis. E4 localizes to the skin, macrophages, dendritic cells in lymph tissue and cartilage. E4 binds to SF proteins like alpha-enolase (ENO1). Effect was epitope-specific and dependent on E4-ENO1 immune complex with FCGR2B on macrophages, which increased IL-10 secretion and reduced osteoclastogenesis. Overall, a well-written paper. Experiment design was well thought out and executed.

1) The authors nicely demonstrate specific binding of E4 to the citrulline sidechain by introducing point mutations, W48M and S51G, in the paratope (called E4m). However, it is unclear why Tryptophan 48 was changed to Methionine, and Serine 51 substituted with Glycine (generally Alanine is used). The reference did not make this clear - E4m was not characterized in the paper that you referenced. Was the purpose of E4m purely to serve as a negative control in subsequent experiments? Clarification is needed in the text (Results).

2) Fig 3. The DAPI staining does not look great (Fig 3.B, and Fig 3.E), especially when compared to the Hoechst stain in the same figure (Fig 4.C).

3) In healthy joints, homeostasis is maintained through multiple mechanisms, including cell differentiation pathways. The authors conclude that E4 suppresses macrophage differentiation to osteoclasts. Did the authors characterize other cell types (i.e. osteoblasts) before and after treatment with E4?

4) The authors conclude that macrophage differentiation into osteoclasts was reduced by E4, while non-differential expansion of mature osteoclasts was observed in the RANKL groups treated with L2 or E4m. Did E4 deplete the pool of macrophages (which resulted in less osteoclast differentiation)? Or, did the number of macrophages remain constant over time in the E4 treated group?

5) Formation of Immune complexes is an interesting topic in autoimmune diseases. Is there an inability to degrade immune complexes (do they accumulate over time?). Suggest expanding on this in the Discussion.

Minor comment:

Consider not using the word "promiscuous" to describe non-specific binding.

RESPONSE TO REVIEWERS' COMMENTS

Authors: First, we would like to first express our gratitude to all reviewers for carefully examining and improving our work. We have now addressed all questions/comments by both the editor and reviewers, including the addition of new confirmatory experiments as suggested. Changes are detailed below and highlighted in the manuscript by **yellow color**. We hope that the improved manuscript could now meet the expectations from the reviewers and editor.

Reviewer #1 (expertise in inflammation and macrophages in rheumatoid arthritis):

In this manuscript a number of chimeric antibodies or mutants were developed from the PB of patients with ACPA+ RA, which contain a human variable domain and a mouse IgG2b constant domain. Also employed were mouse hybridomas that bound citCol2 (IgG2b or IgG1) and a murine arthritogenic antibody to Col2. One of these chimeric antibodies, E4, was observed to suppress CAIA, bind also to ENO1-3 and CitENO1. E4-CitENO1 complexes were able to induce IL-10 more avidly through FCGR2B, and when injected into mice were able to suppress CAIA and GPI induced arthritis, but not other forms of inflammation. E4 was able to bind LPS stimulated macrophages to the same degree as L2, only E4 was effective at decreasing RANKL induced osteoclastogenesis *in vitro*. L2 appeared less effective than E4 at binding RA cartilage and CAIA cartilage, even though it had binds to more citrullinated epitopes.

The observation that ACPA may reduce inflammation is novel and potentially important. However, there are a number of concerns with the manuscript as presented.

1) the notion that there is no data supporting the concept that ACPA may contribute to disease pathogenesis is not correct. There is substantial evidence that ACPA may contribute to disease pathogenesis, however its role is not conclusive.

Authors: We agree with the reviewer's point that there are many reports suggesting pathogenic effects although no firm conclusions can yet be made, and we have now improved the text accordingly (line 67-71 and 297-298). In fact, we had already mentioned in the Introduction and Discussion sections that based on previous analysis ACPAs were postulated to play a role in RA pathogenesis not only because of its association to severe RA but also based on *in vitro* pathogenic evidence. However, the functions of classical "promiscuous" ACPAs *in vivo* are still unclear, although admittedly there are some published experiments, but it might require too large space to explain basic experimental shortcomings. On the other hand, our findings provide a conclusive clue that the nature of at least some ACPAs is not pathogenic, but rather has diverse potentials such as playing a protective role *in vivo*.

2) The conclusions are based on only 1 chimeric antibody. It is difficult to generalize based on an n =1.

Authors: Yes, among the tested monoclonal ACPAs, E4 was the only one with protective effect. On the other hand, ACPA with the same specificity as E4 is common in RA sera so it is likely, but not proven, to be a common phenomenon. We do think it is important to investigate a protective effect in depth and with a full set of controls, involving control mutants. The reason is that it is common that antibodies, in particular recombinant antibodies, could give adverse effects after injection *in vivo*, due to physical effects such as formation of immune complexes or aggregates already before injections. Therefore, such experiments need to be carefully controlled and performed at this stage to screen large number of monoclonal antibodies for protective effects. Nevertheless, this will be done, both by us and likely also by others.

3) The effects of this antibody are based on a mouse IgG constant region, which is necessary for the experiments described to be performed. However, since Fc glycosylation and sialylation are important in the pathogenic potential of ACPA, it is a concern that some

confirmatory experiments using an antibody with a human Fc component were not performed.

Authors: We have now followed the suggestion from the reviewer and expressed the E4 hlgG1 antibody. Since it is not ideal to test the human IgG1 in CAIA because of the isotype origin, we isolated the CD14⁺CD16⁻ monocytes from human PBMCs and differentiated them into macrophages, which allowed us to test the effect of E4 hlgG1 in complex with citENO1 in a setup similar to what we showed in mouse macrophages. We observed similarly increased IL-10 by E4-citENO1 immune complex on human macrophages, but not with unmodified ENO1, again supporting that the effect is dependent on IC formation. The results have now been added to Figure 6I (also line 284-290).

Regarding Fc glycosylation, we think it is still far from being conclusive that Fc galactosylation and sialylation play a pathogenic role. It is however not the topic here as we control for this.

4) Does E4 interfere with the binding of M2139 to cartilage?

Authors: We would like to clarify that the M2139 antibody only recognizes unmodified triple helical J1 epitope of COL2 whereas E4 ACPA does not bind to unmodified or citrullinated J1 epitope in either cyclic or triple helical forms, their binding specificities are completely different without overlapping, therefore it is highly unlikely that E4 could interfere with the binding of M2139 to cartilage.

5) In panel A of Figure 1 the identification of what the symbols represent is unclear, primarily because of the placement of LPS.

Authors: We assumed the reviewer's comment is related to Figure 2A, and we have now improved positioning of the symbols and text in the plot (Fig.2A), we hope they are now clearer now.

6) Figure 1 E should include the quantification of the histologic abnormalities: inflammation etc.

Authors: We assumed the reviewer's comment is related to Figure 2E, and we have now included a histologic scoring figure as suggested (Fig.2F).

7) Figures 3 A-B – clarify that mice were induced with CAIA or not and then the cartilage stained with E4 etc. Also, if the cartilage was stained after CAIA from mice not treated, why would the cartilage from the FCGR2B deficient mice not stain with E4?

Authors: We have now highlighted the descriptive texts for clarifying the reviewer's concerns (line 175-181). Regarding the second question, the FCGR2B KO mice in Fig.3B were not for interpreting the FCGR2B dependence of E4 but rather a model for a robust onset of CAIA and early citrullination in cartilage, as these mice are highly susceptible to CAIA even without the stimulation of LPS. We took the arthritic cartilages from FCGR2B KO mice after day 1 of CAIA induction and they were actually stained by E4 in the superficial zone as a thin layer of staining (Fig.3B, image quality now improved, white arrows), it was not a strong binding because in this early phase of CAIA (<24 hours) we expected a certain rather than a high degree of citrullination in the cartilage, and E4 could already interact with it. We clearly showed that if the CAIA cartilages were from day 15 after CAIA, the staining was stronger (Fig.3A), indicating a more citrullinated cartilage was present.

8) Is Figure 3C is not clear to me. What is represented by ACPA? If this is supposed to show co-expression, I could not see it.

Authors: The original green column title "ACPA" represented the staining of E4, E4m and L2 (row titles) visualized by anti-mouse IgG-CF488 secondary antibody to show the binding to the ATDC5 culture, it has now been changed to "mIgG" for better interpretation. The biotinylated ACC4 was added after the anti-mouse IgG-CF488 staining and visualized by streptavidin-PE to identify the citrullinated/denatured COL2.

9) The inhibition of CAIA seems much less in Figure 5A compared to earlier figures. How variable is the suppression.

Authors: In Figure 5A the arthritis severity was higher than earlier figures, this is expected due to the different batches of arthritogenic cocktails and mice age etc. The effect of E4 is clearly profound in mild or moderate CAIA (mean arthritis score <30 in control group) as in earlier figures (Fig.2C-D), but seemed not as pronounced in more severe arthritis (mean arthritis score >30 in control group) (Fig.4A), although the effect is still significant.

10) In Figure 5E, I do not appreciate the co-expression of F4/80 and ACPA, if that is the purpose.

Authors: We would like to clarify that the purpose of F4/80 staining here was rather for identifying the differentiated macrophages in order to show that ACPAs are binding to the macrophage surface.

11) Figure 4A and C, might be more clear if a more high power view of the relevant data were included.

Authors: We have now followed the suggestion by the reviewer and improved the quality of the images. We also highlighted the binding of E4 in Figure 4C for better inspection.

12) Figure 6 C and E, are very difficult for me to read even when magnified. The color combination does not show well, but as best I can see the data, it seems to be repetitious of panel B.

Authors: We have now enhanced the colors and enlarged the text size in these figures. To clarify, Figure 6A-C are the results from mouse macrophages and Figure 6D-E are the results from RA synovial fluid. We hope the figure is now clearer.

13) Figure 6F is not clear to me. The title and legends suggests that the read out is antibodies binding to cit-ENO1? Maybe the title to the panel should read binding to ENO1 or citENO1?

Authors: We have now changed the title of the panel by following the reviewer's suggestion.

14) Figure S5. The point is not clear nor is the data convincing. Is the binding thought to be through Fc or Fab binding. In panel A, I don't see co-expression and panel B the staining for CD11c and CD11b should demonstrate a negative population.

Authors: We highly appreciate the reviewer for pointing this out. We have now improved the image quality for panel A and the co-localization is clearer. For panel B, we agree with the reviewer and re-analyzed the original data with more optimized gating strategies, this time we also looked at the markers such as F4/80, Ly6G and Ly6C. The E4 antibody was indeed binding to the spleen macrophages as well as dendritic cells *in vivo*, the binding to macrophages was predominant and to DCs was less but still significant. The re-analyzed data provides a more solid evidence to support our interpretation regarding the interaction between E4 and macrophages. We decided to move these results from the supplementary materials to the main figure including the full details (Fig.4E-G). We hope the new panel is more convincing now. Regarding the binding part of E4, we clearly showed that the binding of E4 to macrophages is mainly dependent on its immune complex (Fc part) interacting with Fc receptor (Fig.5A-E), the immune complex may form in different ways even under normal conditions with physiologically citrullinated antigens, which is a classic property of ACPA echoing with the staining in naïve tissues (Fig.4A-D), whereas M2139 is highly specific to native COL2 in the joint, explaining why M2139 does not bind to splenic macrophages like E4 (Fig.4E-G).

15) Figure S7: the macrophages may already have been maximally stimulated to produce TNFa. Do the complexes induce TNFa in macrophages not stimulated with LPS or with a suboptimal dose. A positive control to show that it is possible to induce more TNF by other complexes should be included.

Authors: As the reviewer points out, we indeed expected higher levels of TNF-alpha after the overnight stimulation by LPS in this experiment. However, we would like to clarify that the LPS stimulation was combined with ACPA ICs, therefore the increased cytokine level was caused by the combined effect by LPS and ACPA ICs. Since we did not see any significant change when comparing LPS + ACPA ICs to LPS control group, we concluded that the addition of ACPA ICs did not enhance or contribute to the promotion of TNF-alpha in this experiment. To better address the concerns by the reviewer, we have now added a new experiment with similar setups but without LPS stimulation, we also added the pathogenic antibody M2139 in complex with COL2 as a positive control. As a result, we could already observe an increase of TNF-alpha when naïve macrophages were treated by M2139-COL2, though it was not as potent as LPS stimulation. When combining M2139-COL2 with LPS, we could also observe an enhanced secretion of TNF-alpha, such effect was not observed with E4-citENO1 (Fig.S7B).

16) Why E4 and not other antibodies with similar specificities and Fc domains did not demonstrate the reported effect should be more convincingly discussed.

Authors: We appreciate the point from the reviewer and tried to the Discussion section with text below, highlighted in the manuscript (line 351-368).

"Although a clear picture of the role of ENO1 is still lacking, it is intriguing to speculate that citrullinated ENO1 could at least serve as an antigen included in the formation of immune complex together with ACPAs. Emerging evidence suggest that allosteric effects by antigen-binding of IgG has an intrinsic impact on the immune complex conformation and changes the Fc-FcR-mediated effector function⁷¹⁻⁷³. E4-containing ICs interact with FCGR2B, and these ICs could be formed by certain citrullinated antigens (e.g., citENO1), derived from activated macrophages or extracellular synovial surrounding, whereas L2 displays another binding pattern. We show that the epitope-specificity is critical for mediating the effect of ACPAs. However, even though E4 and L2, or the other ACPAs tested in this study might have partially overlapping epitope-specificity in our peptide panel (Fig.1), their binding to many other citrullinated antigens/epitopes such as citENO1, citCOL2, or the ones tested in our recent study¹², could be vastly different, especially due to the dynamic in vivo environment as well as the variety and abundance of citrullinated antigens in RA. The indication from L2, which does not regulate CAIA or macrophages but shares the constant domains with E4, indicates that ACPAs could form immune complexes with high variety, resulting in different downstream effects. ACPAs specifically recognize a citrulline sidechain, but could have a variable binding spectrum, depending on positive or negative interaction with surrounding amino acid side chains¹⁰⁻¹². The interaction between monoclonal ACPA and citrullinated proteins exposed in an in vivo context, which could be changed in an inflammatory site, could determine the formation of different types of immune complexes."

We would also like to highlight that, even though E4 and L2 share a few epitopes in the COL2 peptide panel (Fig.1), E4 is clearly more restricted in this regard. However, unlike L2, E4 does not bind to those COL2 epitopes that are mutually recognized by pathogenic COL2 antibodies such as M2139, ACC2 and ACC4 (line 112-114).

Reviewer #3 (expertise in rheumatoid arthritis and ACPA):

Anti-Citrullinated Protein Antibodies (ACPA) is a hallmark of RA. The authors expressed monoclonal ACPAs from RA patients, in mice. Antibodies did not induce pain or RA. One antibody, E4, protected against antibody-induced arthritis. E4 localize to the skin, macrophages, dendric cells in lymph tissue and cartilage. E4 binds to SF proteins like alpha-enolase (ENO1). Effect was epitope-specific and dependent on E4-ENO1 immune complex with FCGR2B on macrophages, which increased IL-10 secretion and reduced osteoclastogenesis. Overall, a well written paper. Experiment design was well thought out and executed.

1) The authors nicely demonstrate specific binding of E4 to the citrulline sidechain by introducing point mutations, W48M and S51G, in the paratope (called E4m). However, it is unclear why Tryptophan 48 was changed to Methionine, and Serine 51 substituted with Glycine (generally Alanine is used). The reference did not make this clear - E4m was not characterized in the paper that you referenced. Was the purpose of E4m purely to serve as a negative control in subsequent experiments? Clarification is needed in the text (Results).

Authors: We appreciate the reviewer for pointing this out and we have realized that there was an error in the text, the mutation of S51 on E4m should be S51A instead of S51G and it is now corrected and clarified with text highlighted in the manuscript (line 92-97). The introduction of S51A is to understand whether the size of the binding pocket on E4 is important, since by replacing Ser with Ala, the pocket size becomes smaller and the citrulline side chain may not fit well. Regarding W48, in short, we in fact have made two variants, i.e. W48M and W48E, compared their binding to the COL2 peptides (same peptides as Figure 1) and could not observe any binding from neither of them, we have now added the results in the manuscript as suggested (Supplementary Fig.1B). The W48E is to introduce negative charge to the binding pocket, we expected this variant could therefore make it possible for E4 to bind to positively charged arginine instead of citrulline, but it turned out that E4m-W48E just does not bind to any of them, same as E4m-W48M. On the other hand, W48M showed higher yield in our expression system, considering that large scale production was needed for *in vivo* experiments, we finally chose W48M + S51A mutations for E4m antibody. The employment of E4m is to serve as a negative/isotype control in the indicated experiments.

2) Fig 3. The DAPI staining does not look great (Fig 3.B, and Fig 3.E), especially when compared to the Hoechst stain in the same figure (Fig 4.C).

Authors: We have now improved the image quality of the mentioned figures, we hope they are easier to interpret.

3) In healthy joints, homeostasis is maintained through multiple mechanisms, including cell differentiation pathways. The authors conclude that E4 suppresses macrophage differentiation to osteoclasts. Did the authors characterize other cell types (i.e. osteoblasts) before and after treatment with E4?

Authors: We appreciate the reviewer for the question about the role of E4 in different cell types regarding the cartilage homeostasis. In this manuscript, we investigated the effect of E4 antibody by inducing osteoclastogenesis using BMDMs. Unlike osteoblasts or chondrocytes that come from mesenchymal stem cells (MSCs), macrophages come from hematopoietic stem cells (HSCs) and do not differentiate into osteoblast. Importantly, neither MSCs nor osteoblasts normally possess Fc receptors. Since we suggested based on our findings that the function of E4 is closely related to immune complex interacting with FCGR2B, it could be difficult to interpret the effect of E4 on osteoblast differentiation, if any. Alternatively, we looked at the LPS-challenged chondrocytes and could observe an improved viability in these cells by E4 treatment (results attached below). However, despite that chondrocyte is an important component in cartilage homeostasis and have been reported to possess Fc receptors¹, the exact role of these FcγRs in cartilage homeostasis is still of uncertainty, it is again difficult to interpret the mechanism of E4 on chondrocytes, but highly interesting to look at for further investigations.

Viability of ATDC5 chondrocytes in different time points after mAbs treatment and LPS challenge. Cells were cultured for 14 days with supplement of insulin-transferrin-selenium (ITS) following the mAb treatment (20 $\mu\text{g/ml}$) and LPS challenge (500 ng/ml). To observe viability in different time points cells were detached from the plate by StemPro™ Accutase™ Cell Dissociation Reagent (Thermo Fisher Scientific) and stained by trypan blue for counting. Results were analyzed with one-way ANOVA and presented as mean \pm SEM.

4) The authors conclude that macrophage differentiation into osteoclasts was reduced by E4, while non-differential expansion of mature osteoclasts was observed in the RANKL groups treated with L2 or E4m. Did E4 deplete the pool of macrophages (which resulted in less osteoclast differentiation)? Or, did the number of macrophages remain constant over time in the E4 treated group?

Authors: We have counted the total cell numbers and could not observe a depletion of macrophages by E4 (results attached below), therefore we could conclude that the number of macrophages remained constant in the E4 treated groups. However, E4m and L2 treatment did not further increase or reduce the osteoclast number comparing to RANKL control group (Fig.5F-G), hence we speculated that the osteoclastogenesis in these two groups was driven mainly, if not only, by RANKL. Please note that all the groups with RANKL (except for E4 groups) tend to have less macrophages since some of them could have fused into osteoclasts.

Total cell numbers in fields assessed in Figure 5F-G. Results were analyzed with one-way ANOVA and presented as mean \pm SD.

5) Formation of Immune complex's is an interesting topic in autoimmune diseases. Is there an inability to degrade immune complexes (do they accumulate over time?). Suggest expanding on this in the Discussion.

Authors: We have now followed the suggestion from the reviewer and extended the discussion in the manuscript (line 326-328 and 351-368).

Minor comment:

Consider not using the word "promiscuous" to describe non-specific binding.

Authors: We would like to clarify that the expression "promiscuous" is a way to describe hapten specificity and was used decades ago for describing this recognition. In fact, it does not describe multi-reactivity, quite the opposite, it describes citrulline (i.e., here the hapten) specificity. We have earlier described the meaning of promiscuity using E4 as an example². ACPAs like E4 are specific to citrulline residue, but the amino acids neighboring the citrulline residue on the epitope might have a negative impact on the binding. Thus, the binding of different ACPAs to citrullinated epitopes is usually diverse, regardless of mutually being specific to citrulline, hence we adopted the word "promiscuous" which accurately describe the situation². We are aware of that this terminology has not yet been accepted in the rheumatology field, we have therefore now replaced the word "promiscuous" by "citrulline-specific" where this is possible.

Reference

1. Stock, M. *et al.* Fc-gamma receptors are not involved in cartilage damage during experimental osteoarthritis. *Osteoarthr. Cartil.* **23**, 1221–1225 (2015).
2. Ge, C. & Holmdahl, R. The structure, specificity and function of anti-citrullinated protein antibodies. *Nat. Rev. Rheumatol.* **15**, 503–508 (2019).

REVIEWERS' COMMENTS

Reviewer #1 (expert in inflammation and macrophages in RA):

My concerns have been adequately addressed.

Reviewer #2 (expert in ACPA, rheumatoid arthritis):

The authors have addressed my comments